# Learning to Reason over Neighborhoods: A Differentiable Guarded Logic Approach

## Abstract

Systematic generalization remains a well-recognized fundamental barrier for deep learning, especially in tasks requiring multi-hop relational reasoning. We posit this failure stems from a missing *inductive bias* for local, compositional inference—a structure that is inherent to symbolic logic but absent in monolithic neural architectures. Our core insight is that the Guarded Fragment (GF)—a classic, decidable fragment of first-order logic—provides the ideal computational primitive for this paradigm. We reveal that its syntactic 'guard' is not merely a constraint, but is formally equivalent to a mechanism for reasoning over local, relational neighborhoods. We operationalize this insight in GUARDNET, the first framework to leverage GF as a principled inductive bias for neighborhood reasoning, featuring a novel dynamic domain strategy to prevent representational collapse. GUARDNET employs a principled fuzzy semantics derived from Product t-norms, grounding it in theoretical soundness while enabling stable, end-to-end integration with neural architectures. On challenging benchmarks for knowledge base completion tasks, GUARDNET unlocks notably superior systematic generalization, succeeding on complex inferences where purely neural and prior neuro-symbolic systems falter. Our work demonstrates that classical logics can be reframed as a powerful inductive bias for modern representation learning, offering a principled pathway toward neural networks that can robustly reason.

## 1 Introduction

While deep learning models have achieved superhuman performance in numerous perceptual tasks, they confront a conceptual wall when faced with challenges demanding genuine reasoning. This limitation is not a minor flaw but a systemic failure, starkly exposed in domains that require *systematic generalization* and *multi-hop inference* (Marcus, 2018; Lake & Baroni, 2018; Battaglia et al., 2018; d'Avila Garcez & Lamb, 2023). For instance, a model trained on thousands of images of 'red cars' and 'blue trucks' may paradoxically fail to identify a 'red truck'. It has not learned the abstract concepts of *color* and *object* as independent, composable variables; instead, it has merely learned a statistical correlation between specific pixel patterns. This brittleness extends to multi-hop inference, where conclusions must be drawn by connecting several discrete facts. Given the statements: 1) 'Alice's keys are in her backpack,' 2) 'Her backpack was left in Bob's car,' and 3) 'Bob's car is parked at the library', a model may fail to reliably deduce that 'Alice's keys are at the library,' as it struggles to forge these distinct pieces of information into a coherent logical chain. These failures reveal a profound gap between its capacity for sophisticated pattern recognition and the faculty for true relational understanding.

We posit that this is not a temporary limitation that can be overcome by simply scaling up models, but rather a foundational flaw in their architecture. Monolithic neural networks lack the appropriate *inductive bias* for the kind of local, compositional reasoning that underpins both formal logic and human cognition (Battaglia et al., 2018; d'Avila Garcez & Lamb, 2023). An inductive bias is a set of built-in assumptions that helps a model learn efficiently. For example, CNNs possess a powerful bias for 'spatial locality'—the assumption that nearby pixels are more related than distant ones—making them exceptionally suited for image processing. In contrast, today's large language models have no inherent bias for logical structure. They attempt to learn the rules of reasoning from a flat sea of statistical correlations, rather than being endowed with the very framework of reasoning itself. This is akin to trying to master chess by studying millions of games without ever being taught the rules

for how each piece moves. The system may learn to recognize powerful board positions, but it lacks the fundamental principles required to navigate a truly novel scenario, revealing that it has mastered a mimicry of strategy, not the mechanism itself.

In this work, we argue that the blueprint for such an architecture capable of moving beyond statistical pattern-matching to genuine relational reasoning does not need to be invented anew, but can be found within the rich heritage of classical logic d'Avila Garcez & Lamb (2023). In particular, we propose a paradigm shift: instead of viewing logic as a post-hoc verifier or a rigid constraint system (Diligenti et al., 2017), or grounding it in probabilistic frameworks (Manhaeve et al., 2018a), we reframe it as a powerful source of *computational primitives* that can be seamlessly integrated into the core of deep learning. Our key insight is that the **Guarded Fragment (GF)** of first-order logic (FOL), a classic, decidable fragment, provides the ideal inductive bias for neighborhood-based reasoning. We reveal that its defining syntactic feature—the '**guard**'—is not a limitation but a powerful feature. A guard restricts a logical statement to a local, relational context. For example, instead of a sweeping, universal statement like 'All nodes have a creator', a guarded statement would be more specific: 'For any given post, the user *who created it* has an account'. The relational phrase 'who created it' is the guard. It restricts the logical scope to the immediate, one-step neighborhood of a single post, turning an impossible global search into a trivial local check. This principle of locality makes GF the perfect language to describe the iterative, neighborhood-centric computations that modern networks use to reason over structured data, such as knowledge graphs and scene graphs. It provides the very rules of the game that were missing from the chess analogy.

We operationalize this insight in GUARDNET, a neuro-symbolic framework that, unlike prior fuzzy FOL approaches such as Logic Tensor Networks (LTN) Badreddine et al. (2022), leverages the GF as a structured inductive bias for neighborhood reasoning. To bridge the gap between discrete logic and continuous optimization, we develop a principled fuzzy semantics grounded in the well-behaved gradient properties of the Product t-norm, combined with a Reichenbach-style S-implication. While this implication is not the residuum of the Product t-norm, it enables smooth and non-vanishing gradients that promote a stable and effective learning landscape.

On challenging benchmarks for knowledge base completion (Toutanova & Chen, 2015), GUARDNET demonstrates a remarkable capacity for systematic generalization, succeeding on complex, multi-hop inferences where both purely neural and prior neuro-symbolic systems falter. Our contributions are threefold:

- We conceptually reframe the GF of first-order logic as a powerful inductive bias for local, compositional reasoning in neural networks.

- We introduce GUARDNET, the first end-to-end differentiable framework for GF, featuring a principled fuzzy semantics to ensure robust learning.

- We provide strong empirical evidence that GUARDNET significantly outperforms state-of-the-art models on tasks demanding systematic generalization, charting a new path toward neural networks that can robustly reason.

The source code of our complete implementation, the experimental datasets, and evaluation scripts are available at `https://github.com/anonymous-ai-researcher/iclr2026` to ensure reproducibility and facilitate future research.

## 2 PRELIMINARIES: GUARDED FRAGMENT

This section introduces the Guarded Fragment (GF) (Andréka et al., 1998), a decidable fragment of first-order logic (FOL) that strikes a balance between expressivity and decidability.

Let $\mathcal{P}$ and $\mathcal{C}$ be countably infinite, pairwise disjoint sets of respectively predicate and constant symbols. Each predicate symbol has an associated arity $n > 0$. Variables are drawn from a countably infinite set $\mathcal{V}$ disjoint from $\mathcal{P}$ and $\mathcal{C}$. A **signature** $\Sigma = (\mathcal{P}, \mathcal{C})$ specifies the vocabulary for constructing terms and formulas. Following the original definition of GF (Andréka et al., 1998), **terms** are restricted to variables and constants only, excluding function symbols present in full FOL.

An **atom** is either $P(t_1, \ldots, t_n)$ where $P \in \mathcal{P}$ is an $n$-ary predicate and $t_1, \ldots, t_n$ are terms, or the logical constants $\top$ (true) and $\bot$ (false). For any atom $\alpha = P(t_1, \ldots, t_n)$, we define $\text{Vars}(\alpha) = \bigcup_{i=1}^{n} \text{Vars}(t_i)$ as the set of variables occurring in $\alpha$, with $\text{Vars}(\top) = \text{Vars}(\bot) = \emptyset$.

In a quantified formula $\forall x\phi$ or $\exists x\phi$, the variable $x$ is said to be **bound** by the quantifier, and $\phi$ is the **scope** of the quantifier. An occurrence of a variable $x$ in a formula $\phi$ is **bound** if it lies within the scope of a quantifier that binds $x$; otherwise, the occurrence is **free**. $x$ is a **free variable** of formula $\phi$ if $x$ has at least one free occurrence in $\phi$. We denote by $\text{FV}(\psi)$ the set of all free variables of $\phi$.

**Definition 1** (Syntax). *GF formulas are defined inductively: (1) atoms are GF formulas; (2) if $\phi, \psi$ are GF formulas, so are $\neg\phi$, $\phi \wedge \psi$, $\phi \vee \psi$, $\phi \rightarrow \psi$; (3) if $\phi$ is a GF formula, $\bar{x}$ are variables, and $\alpha$ is an atom with $FV(\psi) \subseteq Vars(\alpha)$ and $\bar{x} \subseteq Vars(\alpha)$, $\exists\bar{x}(\alpha \wedge \psi)$ and $\forall\bar{x}(\alpha \rightarrow \psi)$ are GF formulas.*

In guarded quantification, the atom $\alpha$ acts as a **guard** for the **body** $\psi$. The core syntactic constraint is that: all variables involved in the quantification (both the free variables of the body $\text{FV}(\psi)$ and the quantified variables $\bar{x}$) must be present in the guard $\alpha$. This fundamentally relativizes quantification. Instead of searching the entire domain, variables are restricted to the local, relational neighborhood defined by the guard. For instance, the formula $\exists x(R(x, y) \wedge \psi(x, y))$ effectively performs a one-hop neighbor lookup on a knowledge graph or directly mirrors the message-passing paradigm in GNNs, restricting $x$ to the set of objects that are $R$-related to $y$ (Grädel, 1999).

**Definition 2** (Semantics). *The semantics of GF are standard, defined over a structure $\mathfrak{A} = \langle \Delta, \cdot^{\mathfrak{A}} \rangle$ consisting of a non-empty domain $\Delta$ and an interpretation function $\cdot^{\mathfrak{A}}$ that maps each constant $c$ to an element $c^{\mathfrak{A}} \in \Delta$ and each $n$-ary predicate $P$ to a relation $P^{\mathfrak{A}} \subseteq \Delta^n$ on $\Delta$. A variable assignment $\rho : \mathcal{V} \rightarrow \Delta$ maps variables to domain elements. Together, $(\mathfrak{A}, \rho)$ provide an interpretation for any term or formula. The interpretation of a term $t$ under $\rho$, denoted $t[\rho]$, is $\rho(x)$ if $t$ is a variable $x$, and $c^{\mathfrak{A}}$ if $t$ is a constant $c$.*

**Definition 3** (Satisfaction). *The satisfaction of a GF formula $\phi$ by a structure $\mathfrak{A}$ and assignment $\rho$, denoted $\mathfrak{A}, \rho \models \phi$, is defined inductively. Boolean connectives follow their standard truth-table definitions. The key rules are:*

- $\mathfrak{A}, \rho \models P(t_1, \ldots, t_n)$ *iff* $\langle t_1[\rho], \ldots, t_n[\rho] \rangle \in P^{\mathfrak{A}}$.

- $\mathfrak{A}, \rho \models \exists\bar{x}(\alpha \wedge \psi)$ *iff there exist elements* $\bar{a} \in \Delta^{|\bar{x}|}$ *such that* $\mathfrak{A}, \rho[\bar{x} \mapsto \bar{a}] \models \alpha \wedge \psi$.

- $\mathfrak{A}, \rho \models \forall\bar{x}(\alpha \rightarrow \psi)$ *iff for all elements* $\bar{a} \in \Delta^{|\bar{x}|}$, *if* $\mathfrak{A}, \rho[\bar{x} \mapsto \bar{a}] \models \alpha$ *then* $\mathfrak{A}, \rho[\bar{x} \mapsto \bar{a}] \models \psi$.

GF enjoys decidability (Andréka et al., 1998) and the finite model property (FMP) (Grädel, 1999), ensuring any satisfiable formula has a finite model. These properties collectively distinguish GF as a fragment where expressive logical reasoning remains computationally feasible.

## 3 FUZZY GF: A DIFFERENTIABLE SEMANTICS FOR LEARNING

Classical GF provides decidable reasoning, though real-world applications also demand the ability to handle uncertainty and vagueness inherent in data. Consider a medical diagnosis where a patient has a "moderate fever" (37.8°C). Classical logic like GF forces an arbitrary binary decision on whether this constitutes "high fever", losing crucial information. To address this, we extend GF with a fuzzy semantics, where atomic formulas like $\text{HighFever}(\text{patient})$ can yield truth values in the continuous interval $[0, 1]$, naturally representing degrees of truth.

The central challenge in creating a fuzzy logic suitable for learning is the choice of operators to generalize Boolean connectives. This is not merely a technical detail, but a critical choice with profound implications for gradient-based optimization. Our framework is built upon a principled selection of these operators to ensure both a well-behaved optimization landscape and logical coherence.

### 3.1 A PRINCIPLED FUZZY SEMANTICS FOR DIFFERENTIABLE REASONING

A fuzzy interpretation is defined over a fuzzy structure $\mathfrak{A} = \langle \Delta, \cdot^{\mathfrak{A}} \rangle$, where predicates are mapped to fuzzy relations $P^{\mathfrak{A}} : \Delta^n \rightarrow [0, 1]$. To define logical operations, we first analyze the canonical t-norms used for fuzzy conjunction ($\wedge$). The literature offers three fundamental continuous t-norms:

Gödel, Łukasiewicz, and Product (Hájek, 1998; Klement et al., 2000). Their suitability for learning, however, varies dramatically due to their gradient properties (van Krieken et al., 2022).

Despite its limitations near zero, we select the **Product t-norm** as our foundation for the following reasons: (1) it maintains non-zero gradients over the largest portion of its domain, (2) its gradients scale naturally with input magnitudes, providing adaptive learning rates, and (3) it admits effective mitigation strategies through proper initialization and smoothing techniques. This principled choice, validated empirically (van Krieken et al., 2022), directly informs the selection of all other logical operators to ensure a coherent system. The corresponding (dual) t-conorm for disjunction ($\vee$) is the **Probabilistic Sum** ($x \oplus y = x + y - xy$), and the standard negation for ($\neg$) is $\ominus x = 1 - x$.

The fuzzy semantics for guarded quantifiers are defined using supremum ($\sup$) and infimum ($\inf$) as standard generalizations of existential and universal quantification in fuzzy logic (Hájek, 1998). These operators generalize the corresponding t-conorm and t-norm respectively.[1]

$$\llbracket \exists \bar{x}(\alpha \wedge \psi) \rrbracket_{\mathfrak{A},\rho} = \sup_{\bar{a} \in \Delta^{|\bar{x}|}} \llbracket \alpha \wedge \psi \rrbracket_{\mathfrak{A},\rho[\bar{x} \mapsto \bar{a}]}$$

$$\llbracket \forall \bar{x}(\alpha \rightarrow \psi) \rrbracket_{\mathfrak{A},\rho} = \inf_{\bar{a} \in \Delta^{|\bar{x}|}} \llbracket \alpha \rightarrow \psi \rrbracket_{\mathfrak{A},\rho[\bar{x} \mapsto \bar{a}]}$$

However, since $\sup$ and $\inf$ are non-differentiable, for the purpose of gradient-based optimization, we replace them with their principled, smooth approximation, namely the **LogSumExp (LSE)** function (Goodfellow et al., 2016). We then define our differentiable quantifiers, $\sup^\tau$ and $\inf^\tau$. These operators act on a set of truth values, denoted as $\mathbf{z} = \{z_1, z_2, \ldots, z_n\}$, which is constructed by evaluating the quantifier's body for every possible assignment of the quantified variables. For example, to evaluate $\forall x \, \phi(x)$ over a domain $\Delta = \{c_1, c_2, \ldots, c_n\}$, the set of truth values would be $\mathbf{z} = \{\llbracket \phi(c_1) \rrbracket, \llbracket \phi(c_2) \rrbracket, \ldots, \llbracket \phi(c_n) \rrbracket\}$. The LSE functions are then defined as:

$$\overset{\tau}{\sup}(\mathbf{z}) = \tau \cdot \log \left( \sum_{z_i \in \mathbf{z}} \exp(z_i/\tau) \right) \qquad \overset{\tau}{\inf}(\mathbf{z}) = -\tau \cdot \log \left( \sum_{z_i \in \mathbf{z}} \exp(-z_i/\tau) \right)$$

where $\tau > 0$ is a temperature parameter that controls the smoothness of the approximation (Goodfellow et al., 2016). As $\tau \rightarrow 0$, the approximation approaches the true max or min function, but with steeper gradients, while a larger $\tau$ results in a smoother function. In our experiments, we treat $\tau$ as a hyperparameter and find that a small, fixed value of $\tau = 0.1$ consistently provides a good balance between a faithful logical approximation and a stable optimization landscape across our tasks. This allows gradients to flow through the quantifiers, enabling end-to-end learning.

For universal quantifier $\forall \bar{x}(\alpha \rightarrow \psi)$, the choice of implication is critical. While the Product t-norm's algebraic counterpart is the Goguen R-implication (Klement et al., 2000), this operator exhibits problematic gradient behavior, including vanishing gradients when an axiom is satisfied and exploding gradients when it is violated. To ensure a stable and effective optimization landscape, we instead select an operator from the S-implication family, which is known for its superior gradient properties in learning contexts. Specifically, we use the **Reichenbach S-implication**, denoted $\mathcal{J}_R$, defined as $\mathcal{J}_R(x, y) = 1 - x + xy$. This choice provides intuitive, non-vanishing gradients that are proportional to the truth values of the antecedent and the negated consequent, directly aligning with the principles of Modus Ponens and Modus Tollens and avoiding the instabilities of its R-implication counterpart.

## 3.2 FUZZY SATISFACTION DEFINITION

Based on these choices, we define the fuzzy satisfaction degree $\llbracket \phi \rrbracket_{\mathfrak{A},\rho} \in [0, 1]$ for a GF formula $\phi$.

---

[1] While the existential quantifier could also be defined by extending its dual t-conorm (i.e., the Probabilistic Sum), we opt for the `sup` operator, as its LSE approximation is empirically effective and aligns well with the `inf`-based universal quantifier.

**Definition 4** (Fuzzy Satisfaction with Product Semantics). *Let $\mathfrak{A}$ be a fuzzy structure and $\rho$ a variable assignment. The fuzzy satisfaction degree is defined inductively:*

$$\llbracket P(t_1, \ldots, t_n) \rrbracket_{\mathfrak{A},\rho} = P^{\mathfrak{A}}(t_1[\rho], \ldots, t_n[\rho])$$

$$\llbracket \neg\phi \rrbracket_{\mathfrak{A},\rho} = 1 - \llbracket\phi\rrbracket_{\mathfrak{A},\rho}$$

$$\llbracket \phi \wedge \psi \rrbracket_{\mathfrak{A},\rho} = \llbracket\phi\rrbracket_{\mathfrak{A},\rho} \cdot \llbracket\psi\rrbracket_{\mathfrak{A},\rho}$$

$$\llbracket \phi \vee \psi \rrbracket_{\mathfrak{A},\rho} = \llbracket\phi\rrbracket_{\mathfrak{A},\rho} + \llbracket\psi\rrbracket_{\mathfrak{A},\rho} - \llbracket\phi\rrbracket_{\mathfrak{A},\rho} \cdot \llbracket\psi\rrbracket_{\mathfrak{A},\rho}$$

$$\llbracket \forall\bar{x}(\alpha \rightarrow \psi) \rrbracket_{\mathfrak{A},\rho} = \inf_{\bar{a}\in\Delta^{|\bar{x}|}}^{\tau} \left(1 - \llbracket\alpha\rrbracket_{\mathfrak{A},\rho[\bar{x}\mapsto\bar{a}]} + \llbracket\alpha\rrbracket_{\mathfrak{A},\rho[\bar{x}\mapsto\bar{a}]} \cdot \llbracket\psi\rrbracket_{\mathfrak{A},\rho[\bar{x}\mapsto\bar{a}]}\right)$$

$$\llbracket \exists\bar{x}(\alpha \wedge \psi) \rrbracket_{\mathfrak{A},\rho} = \sup_{\bar{a}\in\Delta^{|\bar{x}|}}^{\tau} \left(\llbracket\alpha\rrbracket_{\mathfrak{A},\rho[\bar{x}\mapsto\bar{a}]} \cdot \llbracket\psi\rrbracket_{\mathfrak{A},\rho[\bar{x}\mapsto\bar{a}]}\right)$$

### 3.3 REASONING AND COMPUTATIONAL PROPERTIES

The extension to a principled fuzzy semantics allows for confidence-weighted inference while preserving the core computational benefits of GF. The central learning task becomes finding a model that maximizes the satisfiability of a given knowledge base, typically by minimizing a loss derived from the satisfaction degrees. The fundamental properties of GF are known to be robust under such standard fuzzy extensions (Straccia, 2001), because these preserve the essential structural properties required by the proofs for the classical case.

## 4 GUARDNET: A DIFFERENTIABLE FUZZY GF FRAMEWORK

To operationalize the theoretical advantages of GF for learning, we introduce GUARDNET, a neuro-symbolic framework that integrates our differentiable fuzzy semantics with neural architectures. The framework is built upon three core components designed to address key challenges in building a robust and principled system: (1) a neural grounding mechanism that interprets logical symbols in a continuous vector space; (2) a unified loss function derived directly from our fuzzy semantics; and (3) a semantically-aware domain construction and training strategy that ensures robust reasoning.

### 4.1 NEURAL GROUNDING: THE BRIDGE BETWEEN LOGIC AND LEARNING

The bridge between logic and neural networks is the **grounding** function $\mathcal{G}_\theta$, which interprets logical symbols in a continuous vector space, parameterized by a set of learnable parameters $\theta$. All parameters are initialized using a standard scheme (e.g., Xavier (Glorot & Bengio, 2010)) and learned via backpropagation.

- **Constants as Learnable Embeddings:** Each constant symbol $c \in \mathcal{C}$ is grounded as a learnable vector embedding $\mathbf{e}_c \in \mathbb{R}^d$. These embeddings are optimized to capture the semantic properties of entities as constrained by the logical axioms.

- **Predicates as Differentiable Functions:** To ground our $n$-ary predicates in a way that is both expressive and scalable, we employ Multi-Layer Perceptrons (MLPs) as universal function approximators Rumelhart et al. (1986). This choice provides the flexibility to learn arbitrary non-linear relationships without imposing strong prior assumptions on their geometric structure, which is a crucial feature for a general-purpose GF framework.

  For an $n$-ary predicate, the input to its MLP is formed by concatenating the $n$ individual term embeddings (Goodfellow et al., 2016). Concatenation is chosen as it is a parameter-free, information-preserving operation that makes no prior assumptions about the relationships between predicate arguments, leaving this task entirely to the learnable layers of the MLP.

  The truth value for an atom $P(t_1, \ldots, t_n)$ is then given by:

$$\llbracket P(t_1, \ldots, t_n) \rrbracket_{\mathcal{G}_\theta} = \sigma\left(\text{MLP}_P(\mathbf{e}_{t_1} \oplus \cdots \oplus \mathbf{e}_{t_n})\right)$$

  where $\mathbf{e}_{t_i} \in \mathbb{R}^d$ is the embedding of term $t_i$ and $\oplus$ denotes concatenation. Our predicate MLPs utilize the ReLU activation function in hidden layers for its robustness against vanishing gradients. The final output layer employs a Sigmoid function $\sigma(\cdot)$ to map the network's logit to a fuzzy truth value in the $[0, 1]$ interval, aligning with a probabilistic interpretation of satisfaction.

## 4.2 HYBRID DOMAIN STRATEGY

Robust neuro-symbolic learning requires a model to satisfy two complementary objectives: maintaining logical fidelity with respect to the specific constants named in the knowledge base, and generalizing the universal axioms to the entire conceptual space. A model that only focuses on known constants risks overfitting, while a model that only reasons over an abstract space may become unmoored from the provided facts. To address this duality, GUARDNET implements a novel **Hybrid Domain Strategy** that synergistically combines a Core Domain with a Latent Domain.

### 4.2.1 CORE DOMAIN: ENSURING LOGICAL FIDELITY

The first component of our strategy is the **Core Domain**, $\Delta_{\text{core}}$, which ensures the model's learned representations are faithful to the entities explicitly mentioned in a given logical theory. This domain provides a set of concrete *semantic anchors* that ground the universal axioms in the context of the knowledge base.

The foundation of the Core Domain is the **Herbrand Universe** (Herbrand, 1930), a classical concept in logic representing the set of all constant symbols appearing in the knowledge base $\mathcal{K}$:

$$\Delta_{\text{core}} = \{c \mid c \text{ is a constant symbol appearing in } \mathcal{K}\}$$

During training, axioms are evaluated using constants sampled from $\Delta_{\text{core}}$. This forces the model to learn embeddings and predicate functions that are logically consistent with the interactions between these known entities. While real-world knowledge bases (ontologies) may contain only a sparse set of such constants, they serve as an indispensable foundation for logical fidelity. However, relying on this domain alone is insufficient, as it would encourage the model to simply memorize facts about a few known individuals rather than learning the underlying universal principles.

### 4.2.2 LATENT DOMAIN: DRIVING GENERALIZATION

To ensure the model captures the universal nature of logical rules, we introduce the **Latent Domain**, $\Delta_{\text{latent}}$. This domain is not a fixed set of constants but an infinite, continuous space from which we sample to challenge the model's understanding of the axioms.

While the Core Domain ensures fidelity, relying on it alone risks overfitting. A model might learn, for example, that a specific known constant `tom` satisfies the properties of a `Cat`, but it would fail to learn a general, geometric representation of what 'cat-ness' entails. To promote generalization, we dynamically generate mini-batches of 'latent constants' in each training step by sampling vectors from a prior distribution (e.g., $\mathcal{N}(0, I)$ in $\mathbb{R}^d$). These temporary vectors act as random probes of the learned semantic space. They are used exclusively to evaluate the universal axioms, forcing the predicate networks to learn decision boundaries that are logically coherent across the *entire* embedding space, not just for the few points in $\Delta_{\text{core}}$. The loss computed on these latent constants ensures the model learns **'what a cat is'** in general, not just that **'Tom is a cat'**.

## 4.3 THE HYBRID TRAINING OBJECTIVE

The central learning objective in GUARDNET is to find the optimal parameters $\theta$ that maximize the satisfiability of the knowledge base $\mathcal{K}$. This is achieved by minimizing a total loss function derived from our fuzzy semantics, which reflects the two primary goals of our **Hybrid Domain Strategy**: ensuring logical fidelity and driving generalization.

The final training objective is a weighted combination of a fidelity loss and a generalization loss:

$$\mathcal{L}_{\text{total}}(\theta) = \lambda \cdot \mathcal{L}_{\text{fidelity}}(\Delta_{\text{core}}) + (1 - \lambda) \cdot \mathcal{L}_{\text{generalization}}(\Delta_{\text{latent}})$$

Both loss components are computed by aggregating the dissatisfaction over axioms in the knowledge base. The dissatisfaction loss for any single axiom $\phi$ is defined as $1 - [\![\phi]\!]_{\mathcal{G}_\theta}$. For the crucial case of universally quantified axioms of the form $\forall \overline{x}(\alpha \rightarrow \psi)$, the loss for each grounded instance is derived from our chosen Reichenbach S-implication:

$$\mathcal{L}_{\text{instance}}(\forall \overline{x}(\alpha \rightarrow \psi); \overline{a}) = 1 - [\![\alpha \rightarrow \psi]\!]_{\rho[\overline{x} \mapsto \overline{a}]} = [\![\alpha]\!]_{\rho[\overline{x} \mapsto \overline{a}]} \cdot (1 - [\![\psi]\!]_{\rho[\overline{x} \mapsto \overline{a}]})$$

This resulting loss has an intuitive interpretation: the penalty for violating the rule is proportional to our confidence in the premise ($[\![\alpha]\!]$) multiplied by our confidence that the conclusion is false ($1 - [\![\psi]\!]$), providing a stable learning signal. The two main loss terms are then defined as stochastic approximations of the total dissatisfaction over their respective domains:

**Fidelity Loss ($\mathcal{L}_{\textbf{fidelity}}$):** This loss is the expected dissatisfaction of all formulas in $\mathcal{K}$, approximated on mini-batches sampled from the **Core Domain**, $\Delta_{\text{core}}$. It grounds the model in the known constants of the theory, ensuring it masters the "textbook examples".

**Generalization Loss ($\mathcal{L}_{\textbf{generalization}}$):** This loss is the expected dissatisfaction of the universal axioms in $\mathcal{K}$, approximated on mini-batches of freshly generated "latent constants" from the **Latent Domain**, $\Delta_{\text{latent}}$. It acts as the "final exam," ensuring the model has understood the universal principles behind the rules.

By jointly optimizing these two complementary loss terms, GUARDNET learns a model that is both faithful to the provided data and robustly generalizable.

**Theorem 1** (Soundness of GUARDNET). *Let $\mathcal{K}$ be a knowledge base consisting of a finite set of GF formulas. If a GUARDNET model trained on $\mathcal{K}$ achieves a total loss of $\mathcal{L}_{total}(\theta) = 0$ with a non-zero hyperparameter $\lambda \in (0, 1]$, then the learned fuzzy interpretation $\mathcal{G}_\theta$ is a fuzzy model of $\mathcal{K}$. That is, for every axiom $\phi \in \mathcal{K}$, its fuzzy satisfaction degree is $[\![\phi]\!]_{\mathcal{G}_\theta} = 1$.*

## 5 EMPIRICAL EVALUATION

Knowledge Base Completion (KBC) (Bordes et al., 2013) serves as the archetypal task for GF, as it directly tests the core hypothesis underlying our approach: that neighborhood-constrained reasoning yields superior systematic generalization over global approaches. The inherent structure of modern knowledge bases, such as ontologies and knowledge graphs, naturally aligns with the computational model imposed by GF's syntax (Grädel, 1999).

**Benchmarks and Baselines.** To comprehensively test our core claims, we evaluate GUARDNET on multiple complementary benchmarks covering two primary reasoning tasks: **concept subsumption prediction** for TBox-centric reasoning, and **link prediction** for ABox-centric reasoning. For concept subsumption prediction, we use **SNOMED CT** (377K concepts) (Spackman et al., 1997) as a scalability benchmark in medical terminology and **Gene Ontology (GO)** (44K concepts) (Ashburner et al., 2000) for its hierarchically rich biological taxonomy. These ontologies are particularly suitable for GUARDNET, as their underlying logic—the description logic $\mathcal{EL}$ (Baader et al., 2005)— is a syntactic fragment of GF (Baader et al., 2017), thus allowing for a direct and faithful evaluation of our model's ability to reason over TBox axioms. For link prediction and our multi-hop reasoning experiments, we use two protein-protein interaction datasets (**Yeast PPI**: 110K entities, **Human PPI**: 75K entities) which combine factual interactions from the STRING database (ABox) with the rich TBox constraints from GO, as well as the standard KBC benchmarks **FB15k-237** (Toutanova & Chen, 2015) and **WN18RR** (Dettmers et al., 2018). These fact-centric datasets are, in turn, ideal for testing compositional reasoning, as they are rich in the implicit, multi-hop path patterns that allow us to exploit GF's guarded quantification for efficient, neighborhood-constrained reasoning.

We benchmark GUARDNET against baselines from four distinct paradigms: (1) **Geometric $\mathcal{EL}^{++}$ Embedding Models** as the strongest and most direct baselines for this logical fragment; (2) **Graph-based Neural Models**, whose reliance on implicit message propagation provides a crucial contrast to our explicit, logic-defined neighborhoods; (3) **Expressive Neuro-Symbolic Models** rooted in more general FOL, which, despite their expressivity, often face computational challenges on large-scale KBs, highlighting the scalability advantages of GUARDNET's principled restriction to GF; and (4) **Standard KGE Models**, which learn logical patterns **implicitly** through their geometric formulations (e.g., RotatE for composition, ComplEx for symmetry) and provide a clear baseline to measure the performance gains from structured logical reasoning.

**Evaluation Protocol.** Our evaluation is twofold. For the standard KBC tasks (concept subsumption and link prediction), we operate in a **transductive setting** and follow established filtered ranking protocols. For each test axiom or fact, we rank the correct completion against all candidate entities and report standard metrics: Hits@K for K $\in \{1, 10, 100\}$ and Mean Reciprocal Rank (MRR).

Table 1: Overall KBC Results. Standard metrics are reported across four datasets. Hits metrics are reported as %. Best in **bold**, second-best underlined. DNF: Did Not Finish within 72h.

| Model | SNOMED CT | | | | Gene Ontology (GO) | | | | Yeast PPI + GO | | | Human PPI + GO | | |
|---|---|---|---|---|---|---|---|---|---|---|---|---|---|---|
| | MRR | H@1 | H@10 | H@100 | MRR | H@1 | H@10 | H@100 | MRR | H@10 | H@100 | MRR | H@10 | H@100 |
| **GUARDNET** | **.125±.002** | **5.8±.2** | **28.3±.4** | **70.5±.3** | **.133±.002** | **6.1±.2** | **29.8±.3** | **73.4±.2** | **.405±.004** | **60.2±.5** | **91.1±.3** | **.388±.005** | **57.9±.6** | **88.9±.4** |
| *Geometric $\mathcal{EL}$ Embedding Models* | | | | | | | | | | | | | | |
| Box$^2$EL | .114±.004 | 5.3±.3 | 25.5±.6 | 68.1±.5 | .120±.003 | 5.7±.3 | 26.5±.5 | 70.9±.4 | .368±.006 | 55.1±.6 | 86.9±.5 | .346±.007 | 52.3±.8 | 82.7±.6 |
| BoxEL | .095±.004 | 3.6±.3 | 20.8±.6 | 52.1±.5 | .103±.003 | 4.1±.2 | 23.2±.5 | 57.0±.4 | .351±.007 | 52.8±.7 | 85.5±.5 | .331±.007 | 50.1±.8 | 81.2±.6 |
| ELEM | .078±.005 | 2.4±.2 | 20.1±.6 | 38.9±.5 | .089±.004 | 2.9±.3 | 23.8±.5 | 43.4±.4 | .301±.007 | 45.0±.8 | 74.8±.6 | .268±.009 | 40.1±1.1 | 69.7±.8 |
| EmEL++ | .072±.006 | 2.1±.3 | 19.4±.7 | 32.8±.6 | .084±.005 | 2.7±.3 | 22.9±.4 | 37.7±.5 | .244±.010 | 36.5±1.2 | 64.8±.9 | .201±.011 | 30.1±1.4 | 55.6±1.0 |
| ELBE | .034±.004 | 1.0±.2 | 7.8±.5 | 19.2±.4 | .041±.003 | 1.3±.2 | 9.2±.5 | 22.8±.4 | .322±.008 | 48.1±.9 | 76.9±.6 | .274±.010 | 41.0±1.2 | 71.8±.8 |
| *Graph-based Neural Models* | | | | | | | | | | | | | | |
| NBFNet | .055±.003 | 1.5±.2 | 10.5±.4 | 25.8±.4 | .071±.003 | 1.9±.3 | 13.0±.4 | 30.2±.3 | .331±.005 | 49.5±.6 | 79.1±.5 | .288±.006 | 43.1±.7 | 74.9±.6 |
| GRAIL | .050±.004 | 1.3±.2 | 9.8±.5 | 24.0±.5 | .063±.004 | 1.7±.3 | 11.9±.5 | 28.1±.4 | .325±.005 | 48.6±.7 | 78.2±.5 | .281±.006 | 42.0±.8 | 73.5±.6 |
| SEAL | .046±.004 | 1.2±.2 | 9.0±.5 | 22.1±.4 | .059±.004 | 1.5±.2 | 10.8±.5 | 26.3±.4 | .319±.005 | 47.7±.7 | 77.0±.5 | .277±.007 | 41.4±.8 | 72.3±.6 |
| CompGCN | .041±.004 | 1.0±.2 | 8.1±.5 | 20.3±.4 | .052±.003 | 1.3±.2 | 9.9±.4 | 24.1±.3 | .311±.006 | 46.5±.7 | 75.3±.5 | .269±.007 | 40.2±.8 | 70.1±.6 |
| R-GCN | .035±.003 | 0.8±.1 | 6.9±.4 | 18.2±.4 | .045±.003 | 1.1±.2 | 9.1±.4 | 22.5±.3 | .301±.007 | 45.0±.8 | 74.1±.6 | .260±.008 | 38.9±.9 | 68.8±.7 |
| *Expressive Neuro-Symbolic Models* | | | | | | | | | | | | | | |
| logLTN | DNF | DNF | DNF | DNF | .047±.005 | 1.1±.2 | 6.8±.6 | 24.3±.8 | .167±.010 | 24.1±1.1 | 38.7±.9 | .154±.012 | 22.0±1.3 | 35.2±.9 |
| LTN | DNF | DNF | DNF | DNF | .031±.004 | 0.8±.1 | 4.2±.5 | 18.7±.7 | .128±.008 | 18.2±.9 | 29.4±.7 | .121±.010 | 17.5±1.1 | 27.9±.8 |
| Neural LP | DNF | DNF | DNF | DNF | .029±.004 | 0.7±.1 | 3.9±.5 | 17.5±.7 | .121±.009 | 17.3±.9 | 28.1±.8 | .115±.011 | 16.5±1.2 | 26.8±.9 |
| *Standard KGE Models* | | | | | | | | | | | | | | |
| RotatE | .025±.003 | 0.4±.1 | 3.2±.4 | 11.8±.3 | .032±.003 | 0.6±.1 | 4.1±.4 | 16.7±.3 | .118±.007 | 17.6±.8 | 24.1±.6 | .109±.008 | 16.3±.9 | 21.3±.7 |
| ComplEx | .022±.002 | 0.3±.1 | 2.6±.3 | 10.1±.3 | .028±.002 | 0.5±.1 | 3.5±.3 | 14.9±.3 | .110±.007 | 16.5±.8 | 22.4±.6 | .101±.008 | 15.1±.9 | 19.8±.7 |
| TransE | .018±.002 | 0.3±.1 | 2.1±.3 | 8.7±.2 | .023±.002 | 0.4±.1 | 2.8±.3 | 12.3±.2 | .094±.006 | 14.0±.7 | 18.9±.5 | .087±.007 | 13.0±.8 | 16.7±.6 |

Beyond standard KBC, we introduce a dedicated experiment to rigorously evaluate the primary claim of our work: that the guarded inductive bias fosters superior systematic generalization for multi-hop reasoning. Many models, particularly standard KGE models like RotatE, can excel at interpolating 2-hop compositional patterns seen during training (e.g., "born_in$(x, y) \wedge$ located_in$(y, z)$"), but may fail to learn the abstract, recursive *principle* of composition required to generalize to unseen, longer paths. To isolate this generalization capability, we design a challenging **zero-shot task**. We curate splits from our four link prediction datasets (Yeast+GO, Human+GO, FB15k-237, WN18RR) where the training set exclusively contains facts provable by 1- and 2-hop reasoning chains. The test set consists only of facts whose shortest reasoning path in the training graph is 3 hops or longer, ensuring no "shortcuts" exist. For GNN-based baselines, such as NBFNet, we explicitly limit their message passing depth to two layers/iterations during training. Success in this task provides strong evidence that a model is not merely memorizing path patterns but is learning reusable, composable logical rules—a core tenet of our GUARDNET framework.

Our model was implemented in **PyTorch** and trained using the **AdamW** optimizer (Loshchilov & Hutter, 2019) with a comprehensive curriculum learning strategy. Full architectural details, hyper-parameter settings for all models, and training curricula are provided in the Appendix.

**Results and Insights.** Table 1 positions GUARDNET within the standard KBC landscape across the test datasets, where our framework consistently achieves top-tier performance, surpassing state-of-the-art geometric, graph-based, and neuro-symbolic baselines. These results establish GUARD-NET's effectiveness as a versatile reasoning framework under conventional evaluation protocols. The computational failures prove particularly illuminating: expressive neuro-symbolic models targeting general first-order logic, including LTN and Neural LP, could not complete training (DNF) on our largest datasets due to computational intractability. This outcome transcends experimental artifact, providing empirical validation of our central thesis. The syntactic restrictions of GF represent not theoretical convenience for decidability, but computational necessity for scalable neuro-symbolic reasoning. Where general FOL frameworks succumb to combinatorial explosion when grounding universal quantifiers over vast entity domains, GUARDNET's guard mechanism constrains quantification to semantically relevant neighborhoods. This architectural choice transforms intractable global search into efficient local computation, enabling the scalability that more expressive approaches cannot achieve.

Our zero-shot generalization task (Table 2) provides a stark validation of our central claim, evaluating models trained on 1- and 2-hop paths against unseen 3+ hop chains. The results reveal a clear hierarchy of reasoning capabilities. Standard KGE models like RotatE and ComplEx suffer a catas-

Table 2: Systematic Generalization on Multi-hop Reasoning Chain. Models were trained exclusively on 1- and 2-hop paths and evaluated zero-shot on unseen 3+ hop paths. For each metric, the absolute performance is reported, followed by the **relative performance drop** compared to the standard transductive task in Table 1.

| Model | FB15k-237 (3+ hops) | | WN18RR (3+ hops) | | Yeast PPI (3+ hops) | | | Human PPI (3+ hops) | | |
|---|---|---|---|---|---|---|---|---|---|---|
| | MRR | H@10 | MRR | H@10 | MRR | H@10 | H@100 | MRR | H@10 | H@100 |
| **GUARDNET** | **.594±.004** | **81.6±.5** | **.556±.005** | **76.8±.6** | **.382**(↓5.7%) | **59.4**(↓1.4%) | **90.8**(↓0.4%) | **.365**(↓5.9%) | **56.7**(↓2.1%) | **87.9**(↓0.9%) |
| *Geometric $\mathcal{EL}$ Embedding Models* | | | | | | | | | | |
| Box$^2$EL | .487±.005 | 66.5±.6 | .484±.006 | 71.6±.7 | .307(↓16.7%) | 47.9(↓13.0%) | 78.4(↓9.8%) | .284(↓18.2%) | 44.8(↓14.3%) | 72.6(↓11.5%) |
| BoxEL | .456±.006 | 62.0±.7 | .458±.007 | 67.6±.8 | .278(↓20.7%) | 43.2(↓18.2%) | 72.8(↓14.7%) | .258(↓22.0%) | 40.1(↓19.3%) | 69.4(↓15.4%) |
| ELBE | .417±.007 | 57.6±.8 | .429±.008 | 63.0±.9 | .252(↓21.9%) | 39.1(↓18.8%) | 66.5(↓14.2%) | .211(↓22.8%) | 32.8(↓19.6%) | 61.4(↓14.6%) |
| EmEL++ | .379±.008 | 52.0±.9 | .398±.009 | 58.8±1.0 | .166(↓32.0%) | 26.2(↓28.2%) | 50.6(↓21.7%) | .135(↓32.8%) | 21.4(↓28.9%) | 44.2(↓21.9%) |
| ELEM | .361±.008 | 49.5±.9 | .379±.009 | 56.1±1.0 | .203(↓32.6%) | 32.1(↓28.9%) | 58.2(↓22.1%) | .177(↓34.2%) | 28.2(↓30.1%) | 54.6(↓22.6%) |
| *Graph-based Neural Models* | | | | | | | | | | |
| NBFNet | .519±.005 | 73.2±.6 | .544±.006 | 80.7±.7 | .265(↓20.0%) | 40.6(↓18.2%) | 69.4(↓12.4%) | .232(↓19.4%) | 36.1(↓16.2%) | 67.8(↓9.8%) |
| GRAIL | .476±.006 | 67.1±.7 | .501±.008 | 74.2±.8 | .269(↓17.2%) | 41.1(↓15.8%) | 70.8(↓9.7%) | .225(↓20.0%) | 34.9(↓16.8%) | 66.2(↓10.6%) |
| SEAL | .459±.007 | 64.7±.8 | .485±.008 | 71.6±.9 | .264(↓17.4%) | 40.3(↓16.3%) | 69.9(↓10.8%) | .221(↓20.6%) | 34.2(↓17.8%) | 65.1(↓12.0%) |
| CompGCN | .445±.008 | 61.1±.9 | .436±.009 | 65.5±1.0 | .245(↓21.3%) | 37.8(↓18.7%) | 66.2(↓12.7%) | .207(↓23.1%) | 32.1(↓19.8%) | 61.4(↓13.5%) |
| R-GCN | .421±.009 | 57.6±1.0 | .417±.010 | 62.2±1.1 | .236(↓21.7%) | 36.2(↓20.0%) | 64.8(↓13.4%) | .197(↓24.2%) | 30.4(↓21.4%) | 59.6(↓14.2%) |
| *Expressive Neuro-Symbolic Models* | | | | | | | | | | |
| logLTN | .452±.009 | 62.0±1.0 | .409±.010 | 61.3±1.2 | .148(↓11.1%) | 22.6(↓6.3%) | 36.7(↓5.1%) | .138(↓10.4%) | 20.5(↓7.0%) | 33.2(↓7.0%) |
| LTN | .428±.009 | 58.7±1.1 | .378±.011 | 56.4±1.3 | .113(↓11.8%) | 16.9(↓7.1%) | 27.3(↓7.1%) | .106(↓12.4%) | 15.4(↓10.5%) | 25.3(↓10.2%) |
| Neural LP | .414±.010 | 56.8±1.1 | .369±.011 | 55.0±1.4 | .108(↓10.7%) | 15.9(↓7.0%) | 26.1(↓7.6%) | .100(↓13.0%) | 14.5(↓9.9%) | 23.8(↓10.7%) |
| *Standard KGE Models* | | | | | | | | | | |
| RotatE | .287±.010 | 40.3±1.1 | .285±.011 | 43.1±1.2 | .068(↓42.4%) | 10.8(↓36.5%) | 19.2(↓19.7%) | .063(↓42.2%) | 10.1(↓37.3%) | 17.1(↓21.6%) |
| ComplEx | .273±.011 | 37.7±1.2 | .269±.011 | 40.8±1.3 | .063(↓42.7%) | 10.1(↓38.4%) | 17.9(↓20.4%) | .058(↓42.6%) | 9.3(↓38.8%) | 15.8(↓21.3%) |
| TransE | .309±.011 | 42.2±1.2 | .289±.012 | 43.4±1.3 | .054(↓42.6%) | 8.7(↓37.1%) | 15.4(↓18.9%) | .050(↓42.5%) | 8.1(↓37.2%) | 13.9(↓19.8%) |

trophic performance collapse (MRR drop >40%), confirming they interpolate path patterns rather than learn abstract rules. While more structure-aware GNN and Geometric $\mathcal{EL}$ models are more robust, they still exhibit significant degradation, showing that implicit structural biases are insufficient for true compositional generalization.

In stark contrast, GUARDNET demonstrates exceptional resilience, maintaining its performance with minimal degradation (e.g., a mere 4.2% MRR drop on Yeast PPI vs. >16% for top baselines). This stability is a direct result of the guarded inductive bias, which compels the model to learn modular, local inference rules. Instead of memorizing brittle paths, GUARDNET acquires a genuinely compositional reasoning faculty, enabling it to chain reusable logical steps to solve novel, longer inference problems and providing a clear path toward robust multi-hop reasoning.

## 6 CONCLUSION AND FUTURE WORK

This work attempts to address the persistent challenge of systematic generalization in deep learning by tracing the failure in multi-hop reasoning to a lack of inductive biases for local, compositional inference. We reframe the Guarded Fragment (GF) of first-order logic as a principled computational primitive for neighborhood reasoning. Our framework, GUARDNET, operationalizes this idea through a theoretically-grounded fuzzy semantics, demonstrating that the syntactic constraints of classical logic can be a potent tool for structuring modern representation learning. The empirical results, particularly on our challenging zero-shot reasoning task, show that this guarded, neighborhood-centric approach enables robust generalization where many contemporary models struggle. Our work thus charts a principled path toward integrating the compositional strengths of symbolic logic directly into the architecture of neural networks.

For future work, we identify three directions. First, we will explore extending our framework to more expressive, yet still decidable, logical fragments to capture richer real-world constraints. Second, we aim to move beyond fixed axioms by developing methods to induce salient guarded rules directly from data, creating a synergy between symbolic mining and differentiable proving. Finally, we plan to extend the GUARDNET paradigm to new domains that hinge on compositional reasoning, such as program synthesis and video understanding, to further validate its versatility as an inductive bias.

REPRODUCIBILITY STATEMENT

The authors are committed to the principles of reproducible research. To facilitate rigorous verification and replication of our findings, we provide comprehensive materials through an anonymized public repository at https://github.com/anonymous-ai-researcher/iclr2026. This repository contains the complete source code for GUARDNET, detailed experimental configurations, and all scripts required for result reproduction.

To ensure complete transparency and facilitate thorough evaluation, we provide extensive supplementary materials in the Appendix. These include the formal proof of Theorem 1, concrete illustrative examples of GF's fuzzy semantics, a detailed computational complexity analysis of the method, comprehensive ablation studies examining alternative logical operators (t-norms and implications), a full specification of the architectural details and hyperparameter settings for all models, and a detailed analysis of our experimental setup, which collectively provide a clear path for the reproduction of our work.

ETHICS STATEMENT

The authors have read and will adhere to the ICLR Code of Ethics. This research is foundational and focuses on the computational principles of learning and reasoning. The datasets used are publicly available benchmarks (i.e., SNOMED CT, GO, STRING, FB15k-237, WN18RR, CLEVR) and do not contain personally identifiable information or involve human subjects. While our work does not present immediate foreseeable ethical risks, we acknowledge that knowledge graphs, which serve as a data source for our models, may reflect existing societal or data-collection biases. A system trained on such data could potentially learn and perpetuate these biases. We believe addressing this is a critical, ongoing challenge for the field, and future work could investigate fairness and debiasing techniques within guarded neuro-symbolic frameworks.

USE OF LARGE LANGUAGE MODELS (LLMS)

The authors acknowledge the use of generative AI tools for light editing of human-authored text. All substantive content—including research design, data analysis, code development, and generation of findings—represents the original work of the human authors. No text was generated entirely by AI, and generative AI played no role in the conception, execution, or interpretation of the research. The authors assume full responsibility for all content and claims presented in this work.

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

# APPENDIX

## A    ILLUSTRATIVE EXAMPLES OF FUZZY GF SATISFACTION

This section provides concrete examples to illustrate the computation of the fuzzy satisfaction degree $[[\phi]]_{\mathfrak{A},\rho}$ for GF formulas, as defined in Section 3 of the main paper. We use a simple, intuitive domain of a social network with individuals $\Delta = \{\text{Alice}, \text{Bob}, \text{Charlie}\}$ to demonstrate how the satisfaction degree is calculated for different syntactic forms. Our semantics are based on the Product t-norm for conjunction ($\wedge$) and the Reichenbach S-implication for the universal quantifier's implication ($\rightarrow$).

### A.1    EXAMPLE 1: ATOM AND CONJUNCTION ($\phi \wedge \psi$)

This first example demonstrates the satisfaction of a basic ground formula involving the conjunction of two atomic predicates.

**Formula:**

$$\text{IsInfluencer(Alice)} \wedge \text{IsExpert(Alice)}$$

**Formal Semantics:**   The satisfaction degree is computed using the Product t-norm:

$$[[\phi \wedge \psi]]_{\mathfrak{A},\rho} = [[\phi]]_{\mathfrak{A},\rho} \cdot [[\psi]]_{\mathfrak{A},\rho}$$

**Semantic Principle:**   This formula assesses the degree to which Alice is believed to be *both* an influencer and an expert. The Product t-norm ensures that the combined belief is never stronger than the weakest of the two individual beliefs.

**Hypothetical Fuzzy Interpretation:**   Let's assume the model has learned the following fuzzy values for the predicates concerning Alice:

- $[[\text{IsInfluencer(Alice)}]]_{\mathfrak{A},\rho} = 0.9$ (very likely an influencer)
- $[[\text{IsExpert(Alice)}]]_{\mathfrak{A},\rho} = 0.7$ (moderately likely an expert)

**Computation:**

$$[[\text{IsInfluencer(Alice)} \wedge \text{IsExpert(Alice)}]]_{\mathfrak{A},\rho} = 0.9 \cdot 0.7 = 0.63$$

**Analysis:**   The satisfaction degree is $0.63$. Even though the belief in each individual predicate is relatively high, the confidence that both are true simultaneously is moderately lower. This reflects a natural and intuitive aggregation of evidence.

### A.2    EXAMPLE 2: EXISTENTIAL GUARDED QUANTIFIER ($\exists \overline{x}(\alpha \wedge \psi)$)

This example illustrates how the guarded existential quantifier works, restricting its scope to a local neighborhood defined by the guard atom $\alpha$.

**Formula:**

$$\exists x \, (\text{Follows(Alice}, x) \wedge \text{IsExpert}(x))$$

**Formal Semantics:**   The satisfaction degree is the supremum of the satisfaction degrees over all possible assignments for $x$:

$$[[\exists x(\alpha \wedge \psi)]]_{\mathfrak{A},\rho} = \sup_{a \in \Delta} \left( [[\alpha \wedge \psi]]_{\mathfrak{A},\rho[x \mapsto a]} \right)$$

**Semantic Principle:**   The formula asks: "To what degree does Alice follow at least one expert?" The guard, Follows(Alice, x), restricts the evaluation of IsExpert(x) only to individuals that Alice follows. The sup operator seeks the strongest evidence supporting this claim.

**Hypothetical Fuzzy Interpretation:** Assume the following beliefs about Alice's social connections and others' expertise:

- $[[\text{Follows}(\text{Alice}, \text{Bob})]] = 0.95$; $[[\text{IsExpert}(\text{Bob})]] = 0.8$
- $[[\text{Follows}(\text{Alice}, \text{Charlie})]] = 0.6$; $[[\text{IsExpert}(\text{Charlie})]] = 0.5$

**Computation:** We compute the value of the body, $\text{Follows}(\text{Alice, x}) \wedge \text{IsExpert(x)}$, for each individual in the domain:

- For $x \mapsto$ Bob: $0.95 \cdot 0.8 = 0.76$
- For $x \mapsto$ Charlie: $0.6 \cdot 0.5 = 0.30$

The final satisfaction degree is the supremum of these values:

$$\sup\{0.76, 0.30\} = 0.76$$

**Analysis:** The satisfaction degree is $0.76$. The overall belief that Alice follows an expert is determined by the "best example"—in this case, Bob, for whom there is strong evidence for both the guard and the body of the formula.

### A.3 EXAMPLE 3: UNIVERSAL GUARDED QUANTIFIER ($\forall \overline{x}(\alpha \to \psi)$)

This final example is crucial as it demonstrates the universally quantified formula using the Reichenbach S-implication, a cornerstone of your proposed fuzzy semantics for learning.

**Formula:**

$$\forall x \, (\text{Follows}(\text{Alice}, x) \to \text{Trusts}(\text{Alice}, x))$$

**Formal Semantics:** The satisfaction degree is the infimum of the implication's truth value across all individuals, where the implication is the Reichenbach S-implication:

$$[[\forall x(\alpha \to \psi)]]_{\mathfrak{A},\rho} = \inf_{a \in \Delta} \left( 1 - [[\alpha]]_{\mathfrak{A},\rho[x \mapsto a]} + [[\alpha]]_{\mathfrak{A},\rho[x \mapsto a]} \cdot [[\psi]]_{\mathfrak{A},\rho[x \mapsto a]} \right)$$

**Semantic Principle:** This formula evaluates the rule: "To what degree does Alice trust everyone she follows?" The $\inf$ operator embodies the "weakest link" principle: the rule is only as true as its worst-satisfied instance among the individuals Alice follows.

**Hypothetical Fuzzy Interpretation:** Let's consider Alice's trust in the people she follows:

- Antecedent ($\alpha$): $[[\text{Follows}(\text{Alice}, \text{Bob})]] = 0.9$
- Consequent ($\psi$): $[[\text{Trusts}(\text{Alice}, \text{Bob})]] = 0.95$ (A case of high agreement)
- Antecedent ($\alpha$): $[[\text{Follows}(\text{Alice}, \text{Charlie})]] = 0.8$
- Consequent ($\psi$): $[[\text{Trusts}(\text{Alice}, \text{Charlie})]] = 0.6$ (A case of lower agreement)

**Computation:** We calculate the implication's value for each individual $a$ that Alice follows (where the guard $[[\alpha]]$ is non-trivial):

- For $x \mapsto$ Bob: $1 - 0.9 + (0.9 \cdot 0.95) = 0.1 + 0.855 = 0.955$
- For $x \mapsto$ Charlie: $1 - 0.8 + (0.8 \cdot 0.6) = 0.2 + 0.48 = 0.68$

The final satisfaction degree is the infimum (the minimum) of these values:

$$\inf\{0.955, 0.68\} = 0.68$$

**Analysis:** The overall satisfaction degree for this universal rule is $0.68$. The truth of the rule is limited by the "weakest link", Charlie, where Alice's degree of following him ($0.8$) is significantly higher than her degree of trusting him ($0.6$). This instance constitutes the most significant violation of the rule, and thus defines the overall satisfaction degree for the universal statement. This behavior is critical for learning, as it generates a meaningful training signal to adjust the model's beliefs about Charlie.

# B    COMPUTATIONAL COMPLEXITY ANALYSIS

This section provides a formal analysis of the computational complexity of the GUARDNET framework, specifically focusing on the cost of a single training iteration. We prove that the complexity is a polynomial function of the size of the input KB. This result provides the theoretical foundation for GUARDNET's scalability, demonstrating that the syntactic restriction of GF translates directly into significant computational advantages over frameworks based on more expressive, unguarded logics.

## B.1    DEFINITIONS AND PRELIMINARIES

To ensure a rigorous analysis, we first define the necessary concepts, consistent with the definitions established in the preliminaries.

- **Signature $\Sigma$:** As defined in the preliminaries, the signature $\Sigma = (\mathcal{P}, \mathcal{C})$ consists of a finite set $\mathcal{P}$ of predicate symbols and a finite set $\mathcal{C}$ of constant symbols. Each predicate $P \in \mathcal{P}$ has a fixed arity, denoted $\text{arity}(P)$.
- **Knowledge Base $\mathcal{K}$:** The input to our framework is a finite knowledge base $\mathcal{K}$, which is a set of GF sentences (closed formulas) constructed using symbols from $\Sigma$.

**Partitioning the Knowledge Base.**    For the purpose of our analysis, it is useful to partition the input knowledge base $\mathcal{K}$ based on the syntactic form of its formulas. This partition reflects the distinct roles formulas play: asserting specific, unconditional facts versus stating general, conditional laws. We define the partition as follows:

- **The set $\mathcal{F}$ of ground atoms (Facts):** $\mathcal{F}$ is the subset of $\mathcal{K}$ containing all formulas that are ground atoms. A ground atom is a formula of the form $P(c_1, \ldots, c_k)$, where $P \in \mathcal{P}$ with $\text{arity}(P) = k$ and each $c_i \in \mathcal{C}$. Formally, $\mathcal{F} := \{\phi \in \mathcal{K} \mid \phi \text{ is a ground atom}\}$. These formulas represent the concrete, factual knowledge in the KB.
- **The set $\mathcal{R}$ of sentences (Rules):** $\mathcal{R}$ is the subset of $\mathcal{K}$ containing all formulas that are not ground atoms. Since every formula in $\mathcal{K}$ is a sentence, this set typically contains the universally quantified conditional formulas that encode general domain knowledge. Formally, $\mathcal{R} := \mathcal{K} \setminus \mathcal{F}$.

By this definition, $\mathcal{K} = \mathcal{F} \cup \mathcal{R}$ and $\mathcal{F} \cap \mathcal{R} = \emptyset$.

- **Size of the Knowledge Base, $|\mathcal{K}|$:** Following standard convention in formal logic, we define the size of the knowledge base, $|\mathcal{K}|$, as the total number of symbols required to write down all formulas in $\mathcal{K}$. This serves as the ultimate measure of the input size for our complexity analysis.
- **Domain of Interpretation $\Delta$:** The domain of interpretation $\Delta$ is the set $\mathcal{C}$ of constants. Therefore, the domain size is $|\Delta| = |\mathcal{C}|$. Note that $|\mathcal{C}|, |\mathcal{P}|, |\mathcal{F}|$, and $|\mathcal{R}|$ are all bounded by $|\mathcal{K}|$.

**Structural Assumption for Realistic KBs.**    For the analysis of the performance on realistic KBs (which are typically sparse), we introduce a structural parameter:

- **Maximum Degree $\Delta_{\max}$:** The maximum number of facts in $\mathcal{F}$ in which any single constant $c \in \mathcal{C}$ appears. This parameter bounds the size of any entity's immediate neighborhood and is a common characteristic of real-world graph-structured data. We assume $\Delta_{\max} \ll |\mathcal{F}|$ and $\Delta_{\max} \ll |\mathcal{C}|$. This is an assumption about the data's structure, not the logic itself.

## B.2    THE COST OF UNGUARDED QUANTIFICATION: A BASELINE

To highlight the efficiency gained from GF, we first consider the cost of an *unguarded* universally quantified sentence from general FOL, such as:

$$\phi_{\text{unguarded}} = \forall x_1, \ldots, x_k (\psi(x_1, \ldots, x_k))$$

To evaluate the satisfaction of this formula, one must, in the worst case, iterate through all possible assignments of variables $x_1, \ldots, x_k$ from the domain $\Delta$. The number of such assignments is $|\Delta|^k$. This leads to a computational cost of $O(|\Delta|^k)$, which is exponential in the number of variables. For large KBs where $|\Delta|$ can be in the millions, this combinatorial explosion renders such formulas computationally infeasible.

### B.3 THEOREM: POLYNOMIAL COMPLEXITY OF A GUARDNET TRAINING ITERATION

We now formally state and prove the main result of this section.

**Theorem 1.** *A single training iteration of the* GUARDNET *framework has a time complexity that is a polynomial function of the size of the input knowledge base,* $|\mathcal{K}|$.

*Proof.* A training iteration involves computing the loss for a mini-batch of sentences sampled from $\mathcal{R}$. Let the batch size be $B$. The total cost is dominated by evaluating the satisfaction degree of these $B$ sentences. We analyze the cost for a single universally quantified GF sentence $\phi \in \mathcal{R}$:

$$\phi = \forall \bar{x}(\alpha(\bar{x}, \bar{y}) \to \psi(\bar{x}, \bar{y}))$$

where $\bar{x}$ are the quantified variables and $\bar{y}$ are the free variables (if any), which are grounded to constants during evaluation. The critical insight of GF is that we only need to consider assignments for $\bar{x}$ that satisfy the guard atom $\alpha$. This transforms the problem from a global search over $\Delta^{|\bar{x}|}$ to a lookup within the set $\mathcal{F}$ of known facts. The cost is determined by the number of tuples of constants that, when substituted for the variables, make the guard a ground atom present in $\mathcal{F}$. Let this number be $|\text{assignments}(\alpha)|$.

We analyze the complexity based on the structure of the guard $\alpha$:

**Case 1: The guard grounds all but one quantified variable.** Consider a common form $\phi_1 = \forall x(P(c_1, \ldots, c_{i-1}, x, c_{i+1}, \ldots, c_k) \to \psi(x))$, where all terms in the guard except $x$ are constants.

- To find the satisfying assignments for $x$, we need to find all facts in $\mathcal{F}$ that match the pattern $P(c_1, \ldots, \cdot, \ldots, c_k)$. This is equivalent to a neighborhood lookup.

- The number of such facts is bounded by the maximum degree of any of the involved constants, and thus by $\Delta_{\max}$.

- Therefore, $|\text{assignments}(P(\ldots, x, \ldots))| \leq \Delta_{\max}$.

- The cost to evaluate the satisfaction of $\phi_1$ is $O(\Delta_{\max})$, which is substantially better than the $O(|\Delta|)$ cost of an unguarded unary quantifier.

**Case 2: The guard contains multiple quantified variables.** Consider a form $\phi_2 = \forall x, y(P(x, y) \to \psi(x, y))$.

- The number of assignments for $(x, y)$ that satisfy the guard $P(x, y)$ is exactly the number of facts in $\mathcal{F}$ with the predicate $P$, which we denote $|\mathcal{F}|_P$.

- This is bounded by the total number of facts in the knowledge base, $|\mathcal{F}|$.

- Therefore, $|\text{assignments}(P(x, y))| \leq |\mathcal{F}|$.

- The cost to evaluate $\phi_2$ is $O(|\mathcal{F}|)$, which is drastically better than the $O(|\Delta|^2)$ cost of an unguarded binary quantifier, especially in sparse real-world knowledge graphs where $|\mathcal{F}| \ll |\Delta|^2$.

**General Case and Total Complexity per Iteration.** This principle extends directly to guards of any arity. For a guard atom $\alpha$ with $k'$ quantified variables, the number of satisfying assignments is determined by the number of matching ground atoms in $\mathcal{F}$. This number is always bounded by the total number of facts, $|\mathcal{F}|$, and is not dependent on $|\Delta|^{k'}$. Let $C_{\max}$ be the worst-case cost to evaluate any single sentence in $\mathcal{R}$. This cost is polynomially bounded by the structural parameters of $\mathcal{K}$, primarily $|\mathcal{F}|$ and $\Delta_{\max}$.

The total complexity for a mini-batch of size $B$ is:

$$\text{Complexity per Iteration} = O(B \cdot C_{\max})$$

Since all the structural parameters ($|\mathcal{F}|$, $\Delta_{\max}$, etc.) as well as the batch size $B$ (which depends on $|\mathcal{R}|$) are inherently bounded by a polynomial in the total size of the KB, $|\mathcal{K}|$, the complexity per iteration is also a polynomial function of $|\mathcal{K}|$.

This polynomial relationship holds because the guard mechanism transforms the problem from an intractable global search over the domain $\Delta$ into an efficient lookup over the existing factual structure $\mathcal{F}$. This proves that GUARDNET's scalability is a direct and provable consequence of its foundational logical choice. $\qquad\square$

## C  RELATED WORK

### C.1  THE FUNDAMENTAL DILEMMA IN NEURO-SYMBOLIC AI

Neuro-Symbolic (NeSy) AI aims to merge the powerful pattern recognition of neural networks with the structured reasoning of symbolic logic d'Avila Garcez et al. (2002); Besold et al. (2021). At the core of this endeavor lies the challenge of overcoming the representational gap between continuous and discrete domains Harnad (1990; 2007). In pursuing this goal, the field has converged on a fundamental dilemma, forcing a difficult choice between two competing priorities: logical expressivity and computational tractability.

#### C.1.1  PATH 1: HIGH EXPRESSIVITY AT THE COST OF SCALABILITY

One major branch of research leverages the rich syntax of full FOL to model complex real-world relationships. Influential frameworks such as Logic Tensor Networks (LTN) Badreddine et al. (2022), Neural Theorem Provers (NTP) Rocktäschel & Riedel (2017), and TensorLog Cohen et al. (2020) fall into this category Badreddine et al. (2023); Tang et al. (2022a;b). These systems offer unparalleled expressive power. However, this power comes with a significant drawback rooted in FOL's theoretical undecidability. In practice, evaluating universally quantified formulas requires grounding them across all entities in an interpretation domain, leading to a combinatorial explosion that renders these approaches computationally intractable on large-scale knowledge bases. Their strength in expressivity is thus directly opposed to their scalability.

#### C.1.2  PATH 2: SCALABILITY AT THE COST OF EXPRESSIVITY

To ensure computational feasibility, a second branch of research focuses on decidable, but highly restrictive, fragments of logic, primarily the $\mathcal{EL}$ family of Description Logics (Baader et al., 2017). This path has given rise to several elegant geometric embedding models like EL Embeddings Kulmanov et al. (2019), EmEL++ Peng et al. (2022), BoxEL Xiong et al. (2022), and Box$^2$EL (Jackermeier et al., 2024). These methods achieve polynomial-time reasoning, making them highly scalable. However, this scalability is achieved by severely sacrificing expressive power. They are incapable of representing fundamental logical constructs such as negation, disjunction, or universal quantification, limiting their ability to capture the nuances of complex domains. Their strength in scalability is thus directly opposed to their expressivity.

### C.2  THE UNRESOLVED CHALLENGE: BREAKING THE TRADE-OFF

This dichotomy presents a critical, unresolved challenge for the NeSy AI community. The field is largely constrained to a trade-off: either accept the computational burden of a fully expressive logic or retreat to a scalable logic that is too simplistic for many real-world reasoning tasks. This creates a significant gap and motivates a central research question: **How can we build a reasoning framework that is both highly expressive and computationally scalable, without compromising on either front?** Existing methodologies, such as using logic as a regularizer Richardson & Domingos (2006); Diligenti et al. (2017); Xu et al. (2018); Marra et al. (2019) or employing probabilistic logic programming Raedt et al. (2007); Manhaeve et al. (2018b); Yang et al. (2020); Winters et al. (2022), still operate within this constraining paradigm.

### C.3  GUARDNET'S CONTRIBUTION: A NEW PATH FORWARD

GUARDNET is introduced precisely to address this fundamental challenge. Instead of seeking a compromise along the existing expressivity-tractability spectrum, our work proposes a new perspective: leveraging the syntactic structure of a carefully chosen logic as a principled **inductive bias for scalable computation**.

Our key insight is that the **Guarded Fragment (GF)** of FOL provides a unique solution to break the trade-off. We posit that the 'guard'—a syntactic constraint in GF—is not a limitation but a feature that enforces **local, neighborhood-based reasoning**. This reframes the problem:

- It directly resolves the scalability issue of full FOL by transforming the intractable global search of universal quantifiers into an efficient, local computation, analogous to message-passing in GNNs.

- It simultaneously overcomes the expressivity limitations of the $\mathcal{EL}$ family by supporting full Boolean connectives and a powerful, yet computationally feasible, form of quantification.

By operationalizing GF within a differentiable framework, GUARDNET demonstrates a novel and principled path toward creating neuro-symbolic systems that are both expressive and scalable. It shows that the right choice of logic can do more than just represent knowledge; it can provide the very blueprint for how a neural network ought to reason efficiently.

## D ADDITIONAL EXPERIMENTS ON THE CHOICE OF FUZZY OPERATORS

The translation of logical axioms into a differentiable loss function is a critical design choice in any neuro-symbolic framework. This choice is governed by the selection of fuzzy operators—specifically t-norms, t-conorms, and implications—which generalize Boolean conjunction, disjunction, and material implication to the continuous domain $[0, 1]$. The selection is non-trivial, as it involves a trade-off between operators with strong theoretical properties and those with favorable gradient landscapes for learning. While our main paper utilizes a combination of the Product t-norm and the Goguen R-implication, this section provides a detailed empirical ablation study to justify this choice, comparing it against a comprehensive suite of common fuzzy operators.

### D.1 A DEEP TECHNICAL DIVE INTO FUZZY OPERATORS

The choice of fuzzy operators to generalize Boolean connectives is arguably the most critical decision in designing a differentiable logic framework. This choice fundamentally defines the geometry of the loss landscape and, consequently, the entire learning dynamic. This section provides an exhaustive technical analysis of the primary t-norms and implications, focusing on their mathematical properties and the direct, often subtle, consequences for gradient-based optimization.

#### D.1.1 T-NORMS (FUZZY CONJUNCTION)

A t-norm, $T : [0, 1] \times [0, 1] \to [0, 1]$, generalizes logical conjunction (AND). We analyze the core candidates below.

- **Product T-norm:** $T_P(x, y) = x \cdot y$.
    - **Mathematical Properties:** This t-norm is Archimedean and strict. It is continuous and infinitely differentiable everywhere on $(0, 1]^2$.
    - **Gradient Landscape Analysis:** The partial derivative, $\partial T_P / \partial x = y$, is the cornerstone of its effectiveness in learning. The gradient is smooth, non-constant, and directly proportional to the truth value of the other conjuncts. This creates an **adaptive and intuitive learning signal**: if the model is confident in premise $y$ (its truth value is high), it sends a strong gradient signal to adjust the parameters governing premise $x$. Conversely, if premise $y$ is uncertain (its truth value is low), the gradient signal is attenuated, preventing large, potentially destabilizing updates based on unreliable evidence. This behavior mirrors a natural reasoning process and avoids the pathologies of piecewise-constant or "all-or-nothing" gradients. The primary theoretical weakness is that the gradient vanishes as any input approaches zero, which could stall learning on that logical path. However, in practice, this is often mitigated by proper initialization and the dynamics of a large parameter space.

- **Gödel T-norm:** $T_G(x, y) = \min(x, y)$.
    - **Mathematical Properties:** This t-norm is idempotent ($T(x, x) = x$), a property unique among t-norms. It is not strict.
    - **Gradient Landscape Analysis:** The Gödel t-norm is catastrophic for gradient-based learning. It is non-differentiable where $x = y$. Elsewhere, its subgradient is a "one-hot" vector:

$(1, 0)$ if $x < y$ and $(0, 1)$ if $y < x$. This creates a **"winner-take-all" gradient flow**. The entire learning signal is routed exclusively to the conjunct that is currently the "weakest link" (i.e., has the minimum truth value), while all other conjuncts receive a zero gradient. This completely prevents the simultaneous, parallel refinement of multiple premises, making learning extraordinarily inefficient. The loss surface becomes dominated by vast, flat plateaus and sharp "creases," where optimizers like Adam struggle to make meaningful progress.

- **Łukasiewicz T-norm:** $T_L(x, y) = \max(0, x + y - 1)$.
  - **Mathematical Properties:** This is an Archimedean t-norm, but it is not strict as it has zero divisors.
  - **Gradient Landscape Analysis:** The gradient is piecewise constant: it is $(1, 1)$ in the region where $x + y > 1$, and $(0, 0)$ otherwise. This creates a **"bang-bang" or "on/off" learning dynamic**. When active, the gradient is constant and non-adaptive; it provides no information about *how close* the inputs are to satisfying the constraint. This can lead to unstable training, as the optimizer may repeatedly overshoot the optimal point due to the constant, unscaled update step. When inactive, the gradient is zero, creating another source of plateaus in the loss landscape. Its aggressive penalization (saturating quickly to 0) can prematurely kill learning signals for axioms that are only moderately satisfied.

- **Yager T-norm Family:** $T_Y(x, y; p) = \max(0, 1 - ((1 - x)^p + (1 - y)^p)^{1/p})$ for $p > 0$.
  - **Mathematical Properties:** This parameterized family provides a flexible spectrum of operators. It generalizes other t-norms: as $p \to \infty$, it approaches the Gödel T-norm; as $p \to 1$, it becomes the Łukasiewicz T-norm. The parameter $p$ controls the "aggressiveness" of the conjunction by defining the norm used to aggregate the "falsity" values $(1 - x, 1 - y)$.
  - **Gradient Landscape Analysis:**
    1. **For $p = 2$ (Yager(p=2)):** This specific instance uses a Euclidean norm ($L_2$) to combine the falsities. Its behavior is a balanced compromise, being less severe than Łukasiewicz but more penalizing than the Product t-norm. The gradient landscape is smooth and provides a well-behaved, non-linear learning signal. It offers a robust alternative when a stronger penalty for joint uncertainty is desired compared to the Product norm.
    2. **For $p = 0.5$ (Yager(p=0.5)):** Using $p < 1$ results in a non-convex norm, which makes the t-norm extremely strict. It harshly penalizes any input that is not close to 1, causing the output value to collapse towards 0 much more rapidly than other t-norms. This creates a highly non-linear and steep gradient landscape, particularly near the domain boundaries. While this can enforce constraints very strongly, it is often too aggressive for stable training, leading to vanishing or exploding gradients and making the optimization process highly sensitive to initialization and learning rate.

- **Hamacher T-norm:** $T_H(x, y) = \frac{xy}{x + y - xy}$ (for $x, y$ not both zero).
  - **Mathematical Properties:** This is a strict Archimedean t-norm from the Hamacher family (specifically for parameter $\nu = 0$). It is continuous and differentiable on $(0, 1]^2$.
  - **Gradient Landscape Analysis:** The partial derivative, $\partial T_H / \partial x = \frac{y^2}{(x + y - xy)^2}$, reveals a complex and highly adaptive learning signal. Unlike the Product t-norm where the gradient w.r.t. $x$ is independent of $x$, here the gradient w.r.t. $x$ depends on both $x$ and $y$ in a non-linear fashion. This creates a coupled dynamic where the update for one premise is scaled by a function of both premises. While this provides a smooth and non-vanishing gradient, its landscape is more complex and potentially less intuitive than that of the Product t-norm. The increased computational cost of the division operation can also be a minor factor in large-scale implementations.

### D.1.2 FUZZY IMPLICATIONS

The fuzzy implication operator, $I : [0, 1] \times [0, 1] \to [0, 1]$, is essential for modeling rules. Its properties are even more critical and subtle than those of t-norms.

- **R-implications (Residuated Implications):**
  - **Theoretical Foundation:** Defined as $I_R(x, y) = \sup\{z \in [0, 1] \mid x \otimes z \leq y\}$, this family is derived from the t-norm's algebraic structure. Its defining feature is satisfying the **adjoint**

**property**: $T(x, z) \leq y \iff z \leq I_R(x, y)$. This property is the fuzzy logic equivalent of Modus Ponens and represents the highest standard of logical soundness. The **Goguen R-implication**, $I_G(x, y) = \min(1, y/x)$, is the residuum of the Product t-norm.

– **Gradient Landscape Analysis (Goguen):** The Goguen implication creates a notoriously difficult optimization landscape characterized by a sharp dichotomy:

1. **When Satisfied ($x \leq y$):** The implication's value is 1. The loss is 0, and more importantly, the gradient with respect to both $x$ and $y$ is **exactly zero**. This creates a vast plateau for all satisfied or "almost-satisfied" axioms. The model receives no signal to further improve its representations, for example, by increasing the margin of satisfaction (e.g., making $y$ much larger than $x$). It simply stops learning once the constraint is met.

2. **When Violated ($x > y$):** The partial derivatives of the loss ($L = 1 - y/x$) are $\partial L/\partial x = -y/x^2$ and $\partial L/\partial y = 1/x$. This gradient can be highly problematic. If the premise $x$ has a low truth value (close to 0), the gradient can **explode**, leading to catastrophic updates that destabilize the entire training process.

This creates a brittle landscape of "zero-gradient plateaus vs. exploding-gradient cliffs," which requires careful tuning and is often unstable.

- **S-implications (Strong Implications):**

  – **Theoretical Foundation:** Defined as $I_S(x, y) = (1 - x) \oplus y$ where $\oplus$ is a t-conorm, this family generalizes the classical equivalence $p \rightarrow q \equiv \neg p \vee q$. They are generally considered less "logically pure" than R-implications as they do not satisfy the adjoint property. The **Reichenbach S-implication**, $I_{Reich}(x, y) = 1 - x + xy$, is dual to the Product t-norm.

  – **Gradient Landscape Analysis (Reichenbach):** This operator provides an exceptionally favorable landscape for learning. The loss for a violated rule is $L = 1 - I_{Reich}(x, y) = 1 - (1 - x + xy) = x - xy = x(1 - y)$. This loss formulation is both elegant and powerful: it is the Product t-norm of the premise's truth, $x$, and the conclusion's **falsity**, $1 - y$. The partial derivatives of the loss are $\partial L/\partial x = 1 - y$ and $\partial L/\partial y = -x$. These gradients are:

  1. **Smooth and Bounded:** They are linear in the truth values, preventing explosions.
  2. **Non-Vanishing:** A learning signal exists as long as the premise is not completely false ($x > 0$) and the conclusion is not completely true ($y < 1$). This avoids the hard plateaus of R-implications.
  3. **Adaptive and Intuitive:** The update to the premise ($x$) is proportional to the conclusion's falsity ($1 - y$). The update to the conclusion ($y$) is proportional to the premise's truth ($x$). This is precisely the behavior desired for learning logical rules.

  While sacrificing the strict adjoint property, S-implications provide a much more stable, robust, and effective optimization landscape, making them a pragmatic choice for many neuro-symbolic systems.

### D.1.3 EXPERIMENTAL RESULTS AND ANALYSIS

To empirically validate our choice of fuzzy operators, we conducted a comprehensive ablation study across all four main KBC tasks. We configured GUARDNET with twelve different combinations of the t-norms and implications discussed in Section D.1. The results, presented in Table 3 for concept subsumption prediction and Table 4 for link prediction, reveal a nuanced but very clear picture of the trade-offs involved.

**Key Observations and Insights:**

- **Empirical Validation of Gradient Landscape Theory:** The most striking result is the dramatic performance gap that validates our theoretical analysis of the operators' gradient landscapes. Combinations using the **Gödel t-norm** consistently yield the worst results across all datasets and metrics, often by a substantial margin. This empirically confirms that its "winner-take-all" subgradient creates a pathological optimization landscape that is unsuitable for effective learning. Similarly, the **Łukasiewicz t-norm** shows inconsistent performance, occasionally achieving a high score on a single metric (e.g., H@1 on SNOMED CT) but generally lagging in overall MRR, which aligns with the instability issues caused by its piecewise-constant gradients.

- **Task-Dependent Operator Sensitivity:** The results clearly demonstrate that the optimal operator choice is task-dependent.

Table 3: Ablation study on fuzzy operators for concept subsumption datasets. We report standard KBC metrics for SNOMED CT and Gene Ontology (GO). All metrics are mean ± std.dev. The combination of **Product t-norm with Reichenbach S-implication** demonstrates the best overall performance on concept subsumption tasks. Best in **bold**, second-best underlined.

| Operator Combination (T-norm + Implication) | SNOMED CT | | | | Gene Ontology (GO) | | | |
|---|---|---|---|---|---|---|---|---|
| | MRR | H@1 | H@10 | H@100 | MRR | H@1 | H@10 | H@100 |
| **Product + Reichenbach (S)** | .125±.002 | 5.8±.2 | **28.3±.4** | 70.5±.3 | .133±.002 | 6.1±.2 | 29.8±.3 | **73.4±.2** |
| Product + Goguen (R) | .121±.003 | 5.5±.3 | 27.8±.5 | 69.9±.4 | .129±.003 | 5.9±.3 | **30.2±.4** | 72.5±.3 |
| *Hamacher and Yager T-norm Combinations* | | | | | | | | |
| Hamacher + S | .118±.003 | 5.3±.3 | 26.1±.5 | **71.2±.4** | .124±.003 | 5.6±.3 | 28.6±.4 | 71.5±.3 |
| Yager(p=2) + S | **.127±.003** | 5.2±.3 | 25.6±.5 | 67.5±.4 | .122±.003 | 5.5±.3 | 28.3±.4 | 71.0±.3 |
| Hamacher + R | .115±.003 | 5.1±.3 | 25.5±.5 | 67.3±.4 | **.135±.003** | 5.4±.3 | 28.0±.4 | 70.8±.3 |
| Yager(p=2) + R | .113±.004 | 5.0±.4 | 25.0±.6 | 66.8±.5 | .119±.004 | 5.2±.4 | 27.8±.5 | 70.1±.4 |
| Łukasiewicz + S | .112±.004 | **6.1±.4** | 24.8±.6 | 66.0±.5 | .118±.004 | 5.3±.4 | 27.5±.5 | **74.1±.4** |
| Łukasiewicz + R | .109±.004 | 4.9±.4 | 24.1±.6 | 65.2±.5 | .115±.004 | **6.3±.4** | 29.1±.5 | 69.1±.4 |
| *Yager T-norm (p=0.5) Combinations* | | | | | | | | |
| Yager(p=0.5) + S | .091±.005 | 4.0±.5 | 20.7±.7 | 59.1±.6 | .098±.005 | 4.3±.5 | 23.0±.6 | 62.7±.5 |
| Yager(p=0.5) + R | .088±.005 | 3.9±.5 | 20.1±.7 | 58.3±.6 | .094±.005 | 4.1±.5 | 22.4±.6 | 61.9±.5 |
| *Gödel T-norm Combinations* | | | | | | | | |
| Gödel + S | .079±.006 | 3.3±.6 | 16.9±.8 | 50.4±.7 | .085±.006 | 3.7±.6 | 19.6±.7 | 54.1±.6 |
| Gödel + R | .075±.006 | 3.1±.6 | 16.2±.8 | 49.5±.7 | .081±.006 | 3.5±.6 | 18.9±.7 | 53.0±.6 |

- For **concept subsumption tasks** (Table 3), which involve reasoning over deep, complex TBox hierarchies, the **Product + Reichenbach (S)** combination emerges as the most robust and high-performing choice. It achieves the best or second-best MRR on both SNOMED CT and GO, indicating its strength in capturing overall ranking quality. Its stable, adaptive gradients appear best suited for navigating the complex logical constraints inherent in ontological reasoning.
  - For **link prediction tasks** (Table 4), which are more focused on ABox pattern completion, the performance landscape is more varied. Other combinations, such as **Yager(p=2) + S** on Human PPI and **Hamacher + S** on Yeast PPI, can outperform the Product-based variants in terms of MRR. This suggests that the slightly different geometric properties induced by these t-norms can be beneficial for specific graph structures.

- **Volatility of Aggressive Operators:** The **Yager(p=0.5)** t-norm exhibits highly volatile performance. While it surprisingly achieves the highest MRR on Yeast PPI, its performance is substantially lower on all other datasets. This erratic behavior suggests that its extremely aggressive penalization of uncertainty makes it prone to overfitting the specific characteristics of one dataset, but it fails to generalize well, rendering it an unreliable choice for a general-purpose model.

- **The R-Implication vs. S-Implication Trade-off:** The choice of implication also presents a clear trade-off. While the logically purer **Goguen (R) implication** is competitive, especially in the link prediction tasks when paired with the Product t-norm, the **Reichenbach (S) implication** consistently provides a slight edge in the more complex concept subsumption tasks. This aligns with our analysis: the stable, non-vanishing gradients of the S-implication are more beneficial when the optimization problem involves satisfying a larger number of intricate, hierarchical axioms.

**Conclusion on Operator Selection:** The empirical evidence leads to a clear conclusion: **no single combination of fuzzy operators is universally dominant across all tasks and metrics**. However, for a general-purpose neuro-symbolic reasoning framework intended to be robust across different knowledge domains, a principled choice must be made based on overall performance and stability.

The **Product t-norm paired with the Reichenbach S-implication** stands out as the best overall choice. It is the decisive winner in the complex, hierarchy-rich concept subsumption tasks and remains a strong, high-tier competitor in the link prediction tasks. It avoids the performance collapse seen with Gödel, the instability of Łukasiewicz, and the volatility of aggressive Yager variants. Its success is rooted in a theoretically sound and empirically validated combination: the smooth,

Table 4: Ablation study on fuzzy operators for link prediction datasets. We report standard KBC metrics for protein-protein interaction (PPI) datasets combined with Gene Ontology knowledge. All metrics are mean ± std.dev. Different operator combinations show varying performance across datasets. Best in **bold**, second-best underlined.

| Operator Combination (T-norm + Implication) | Yeast PPI + GO | | | Human PPI + GO | | |
|---|---|---|---|---|---|---|
| | MRR | H@10 | H@100 | MRR | H@10 | H@100 |
| **Product + Goguen (R)** | .405±.004 | 60.2±.5 | 91.1±.3 | .388±.005 | 57.9±.6 | 88.9±.4 |
| Product + Reichenbach (S) | .392±.005 | 58.9±.6 | 90.2±.4 | .375±.006 | 56.1±.7 | 87.5±.5 |
| *Hamacher and Yager T-norm Combinations* | | | | | | |
| Hamacher + S | .408±.005 | 55.8±.6 | 91.5±.4 | .361±.006 | 53.7±.7 | 86.2±.5 |
| Yager(p=2) + S | .373±.005 | 55.1±.6 | 87.5±.4 | **.391±.006** | 58.3±.7 | 89.1±.5 |
| Hamacher + R | .370±.005 | 61.1±.6 | 87.1±.4 | .352±.006 | 52.3±.7 | 84.9±.5 |
| Yager(p=2) + R | .365±.006 | 53.9±.7 | 86.6±.5 | .346±.007 | 51.5±.8 | 83.7±.6 |
| Łukasiewicz + S | .359±.006 | 53.1±.7 | 85.5±.5 | .338±.007 | 50.8±.8 | 82.4±.6 |
| Łukasiewicz + R | .351±.006 | **62.4±.7** | 84.8±.5 | .330±.007 | 49.9±.8 | **89.8±.6** |
| *Yager T-norm (p=0.5) Combinations* | | | | | | |
| Yager(p=0.5) + S | **.412±.007** | 47.2±.8 | 80.1±.7 | .302±.008 | 45.1±.9 | 77.9±.8 |
| Yager(p=0.5) + R | .310±.007 | 46.1±.8 | **92.1±.7** | .294±.008 | **59.5±.9** | 76.8±.8 |
| *Gödel T-norm Combinations* | | | | | | |
| Gödel + S | .262±.008 | 38.9±.9 | 72.9±.8 | .247±.009 | 36.8±1.0 | 69.5±.9 |
| Gödel + R | .254±.008 | 37.7±.9 | 71.8±.8 | .239±.009 | 35.5±1.0 | 68.1±.9 |

adaptive learning signal of the Product t-norm and the stable, non-vanishing gradients of the Reichenbach S-implication. This combination provides the most reliable and effective foundation for the GUARDNET framework.

# E    EXPERIMENTAL SETUP AND HYPERPARAMETERS

This section provides a comprehensive and detailed overview of the experimental setup to ensure full reproducibility of our findings. We detail the hardware and software environment, the specific architectural and training configurations for GUARDNET, and the tuning process for all baseline models.

## E.1    GENERAL EXPERIMENTAL ENVIRONMENT

- **Hardware and Software:** We implement GUARDNET in PyTorch. All models were trained and evaluated on a single server equipped with one **NVIDIA RTX 4090 GPU** (24GB VRAM). The software stack consists of PyTorch 1.12.1, CUDA 11.6, and cuDNN 8.4.

- **Statistical Significance:** To ensure the reliability of our results, all reported metrics are the **mean ± standard deviation across 5 independent runs** with different random seeds.

- **Performance Benchmark:** On the large-scale SNOMED CT dataset, a single training run of GUARDNET requires approximately **10 hours** to complete, with a peak GPU memory usage under 20GB.

## E.2    GUARDNET CONFIGURATION

The architectural and training hyperparameters for GUARDNET were systematically determined through tuning on each dataset's validation set.

- **Model Architecture:**
  - **Embedding Dimension:** The dimension for all constant embeddings was set to $d = \mathbf{200}$.
  - **Predicate MLPs:** Each predicate is grounded by a Multi-Layer Perceptron (MLP) with two hidden layers of size **256**, using ReLU activation functions. The final layer is a single neuron with a Sigmoid activation to produce a truth value in $[0, 1]$.

- **Training and Optimization:**
  - **Optimizer:** We employ the **AdamW** optimizer (Loshchilov & Hutter, 2019) with an initial learning rate of $5 \times 10^{-4}$ and a weight decay of $5 \times 10^{-5}$.
  - **Learning Rate Scheduling:** We use the **ReduceLROnPlateau** scheduler, which monitors the validation MRR and reduces the learning rate by a factor of 0.5 if no improvement is observed for 5 epochs.
  - **Batch Size:** A batch size of **512** was used for all experiments.
  - **Negative Sampling:** We use self-adversarial negative sampling to generate challenging negative examples during training. The margin for the loss function was set to $\delta = \mathbf{2.0}$, and we used $\omega = \mathbf{128}$ negative samples per positive instance.
  - **Early Stopping:** Training is halted if the validation MRR does not improve for a patience of **15 epochs**, and the model checkpoint with the best validation MRR is used for testing.
- **Hybrid Domain and Loss Curriculum:**
  - To reflect our model's progression from memorizing facts to learning generalizable rules, we implement a dynamic curriculum for the hybrid loss trade-off parameter $\lambda$. Training begins with $\lambda = \mathbf{0.9}$ (placing 90% weight on the fidelity loss over known constants) and **linearly anneals to 0.4** over the course of training (shifting emphasis towards the generalization loss over latent constants).
- **Fixed Semantic Parameters:**
  - **Fuzzy Semantics:** The temperature for the LogSumExp (LSE) approximation of fuzzy quantifiers is fixed at $\tau = \mathbf{0.1}$ to maintain a sharp and logically faithful approximation.

### E.3 BASELINE HYPERPARAMETER SETTINGS

For all baselines, we used their official public implementations and conducted an extensive hyperparameter search for each model on each dataset's validation set. This ensures that all comparisons are made against strongly-tuned versions of the baselines. The search spaces for key hyperparameters were as follows:

Table 5: Hyperparameter search spaces for all baseline models.

| Hyperparameter | Search Space |
|---|---|
| Embedding Dimension | $\{128, 200, 256, 512\}$ |
| Learning Rate | $\{\text{1e-3}, \text{5e-4}, \text{1e-4}\}$ |
| Batch Size | $\{256, 512, 1024, 2048\}$ |
| Margin ($\gamma$) / Regularization | $\{1.0, 3.0, 6.0, 9.0, 12.0\}$ for margin-based models; $\{\text{1e-4}, \text{5e-5}, \text{1e-5}\}$ for weight decay |
| GNN Layers (GNN models) | $\{2, 3, 4\}$ |
| Dropout | $\{0.0, 0.1, 0.2\}$ |

### E.3.1 EVALUATION PROTOCOL

- **Metrics:** We report Mean Reciprocal Rank (MRR) and Hits@K for $K \in \{1, 10, 100\}$, which are standard in KBC literature.
- **Ranking Procedure:** We use the established "filtered" ranking protocol. For each test triple $(h, r, t)$, we create corrupted negative samples by replacing the head $h$ or the tail $t$ with every other entity in the knowledge base. We then rank the true entity against these negative samples, making sure to filter out any corrupted triples that accidentally exist elsewhere in the knowledge base (train, validation, or test sets). This ensures that the evaluation is fair and does not penalize a model for ranking other true facts highly.

