# OpenReview forum: "Learning to Reason over Neighborhoods: A Differentiable Guarded Logic Approach"
_ICLR.cc/2026/Conference — ICLR 2026 Conference Withdrawn Submission_

### Official Review · Reviewer_6ReF · 2025-10-21

**Soundness:** 3
**Presentation:** 1
**Contribution:** 1
**Rating:** 2
**Confidence:** 4

**Summary:**

The paper proposes the use of guarded fragment logic for multi-hop reasoning in knowledge bases.

**Strengths:**

- Interesting idea of using guarded fragment of first-order logic
- Outperforms all baselines

**Weaknesses:**

- A dedicated related work section is missing.
- Highly relevant related work is missing, for example [1] and [2].
- Important baselines are missing, e.g., CQD [1].
- In the paper, for a long time, it does not become clear what exactly the task(s) to be solved are. Only at the beginning of the evaluation section, it becomes clear that the paper targets concept subsumption prediction and (multi-hop) link prediction. This should become clear already in the introduction.
- The novelty does not become sufficiently clear. Large parts of Sections 2, 3, and 4 do not seem to be novel (e.g. guarded fragment, fuzzy logic, ...). It should be made more clear what parts are novel.
- The loss function might be new. However, a precise comparison and possibly even an ablation study to previous loss functions would be interesting (e.g., of Box2EL and similar approaches)

[1] Arakelyan, Erik, Daniel Daza, Pasquale Minervini, and Michael Cochez. "Complex query answering with neural link predictors." arXiv preprint arXiv:2011.03459 (2020).

[2] Ren, Hongyu, Mikhail Galkin, Michael Cochez, Zhaocheng Zhu, and Jure Leskovec. "Neural graph reasoning: Complex logical query answering meets graph databases." arXiv preprint arXiv:2303.14617 (2023).

**Questions:**

- How is your approach related to knowledge graph embeddings?
- How is your approach related to neural graph databases [2]?
- How is your approach related to CQD [1]?
- Can you provide evaluation results comparing your approach to CQD?
- What exactly is novel in Section 2 and 3 of the paper?
- What exactly is novel in Section 4 of the paper?

---

### Official Review · Reviewer_ZtWL · 2025-10-28

**Soundness:** 1
**Presentation:** 2
**Contribution:** 1
**Rating:** 2
**Confidence:** 5

**Summary:**

This paper proposes a novel neural reasoning method, ‘reason over neighbourhoods’, for a decidable fragment of first-order logic – Guarded Fragment. An end-to-end differentiable framework – GuardNet -- is implemented for a fuzzy semantics of Guarded Fragment. GuardNet is experimented on several knowledge graphs in two tasks: concept subsumption prediction and link prediction.

**Strengths:**

Authors target to move beyond the statistical paradigm of machine learning, and propose a paradigm shift to view logic as a computational primitive and seamlessly integrated into the core of deep learning.

Using fuzzy logic to reason with knowledge graphs that contain inconsistent information.

**Weaknesses:**

Authors started with a brave claim for a paradigm shift through introducing Guarded Fragment – a fragment of first-order logic – into deep learning, then moved backwards to fuzzy Guarded Fragment and implemented a GuardNet within the statistical paradigm.

The theoretical part, which glowed with crisp logic, became an unreachable utopia in GuardNet with fuzzy logic.

Guarded Fragment can be easily implemented by extending Sphere Neural Networks and keep the rigour of classic logic in the neural world.

The second part (fuzzy logic + GuardNet) can be another paper.

GuardNet will not achieve the total loss of zero, because knowledge base are incomplete and inconsistent. Thus, Theorem 1 is dispensable.

**Questions:**

In the experiment, why didn't you list H@1 results for Yeast PPI+GO, Human PPI + GO, FB15k-237, and WN18RR?

---

### Official Review · Reviewer_VECM · 2025-10-28

**Soundness:** 1
**Presentation:** 2
**Contribution:** 1
**Rating:** 2
**Confidence:** 5

**Summary:**

The authors limit the expressivity of neurosymbolic methods based on fuzzy logic (eg LTN) to the guarded fragment of First Order Logic. They also design a special aggregation function for universal and existential aggregation.

**Strengths:**

- Provides a formal assessment of the complexity benefits of the guarded fragment compared to unrestricted FOL, which is not yet discussed in the neurosymbolic literature.
- Experiments suggest the method gets good performance
- The paper was relatively easy to follow

**Weaknesses:**

- The ideas do not seem novel to me. The LTN framework [1] already has (optional) guarded quantifiers. Except for the different aggregation function, it is not obvious how this paper extends or improves on LTN.
- The paper is quite unclear about its goals. The abstract and introduction mostly talk about multi-hop reasoning and inductive biases, but in the end the paper is about adding a loss function based on background knowledge. This seemed incongruent.
- The LSE aggregator is not properly defined for fuzzy logics
- The experimental description is very unclear

**Questions:**

- How is this paper supporting the claim (abstract) that GuardNet 'reframes logic as a powerful inductive bias for modern representation learning, offering a principled pathway toward neural networks that can robustly reason'? And how is this different compared to existing approaches (eg LTN).
- Since LTN already implements guarded quantification, I am surprised it times out on one dataset where GuardNet doesn't (Table 1). Their complexities should be the same. Why is this?
- On other datasets, how do you explain the large difference in performance with LTN?
- The LSE aggregation functions do not return a value in $[0, 1]$. Eg, say $\mathbf{z}=[0.1, ..., 0.1]\in [0, 1]^{10}$, then the function $f(\tau) = \tau \log (\sum_{i} \exp(z_i/\tau))$ is just a linear function and could return values above 1 depending on $\tau$. Similarly the $\inf$ generalisation would be negative. It is unclear how to interpret these from a fuzzy perspecitve.
    - Note that the paper does define the semantics $[[\phi]]$ as being in $[0, 1]$ so this is definitely not intentional. The soundness also relies on this.
- Experiments:
    - It is not clear where the knowledge comes from that is used to create the theories.
    - It is not clear how the multi-hop reasoning experiments are done
- Related work:
    - This should be in the main text
    - C.1.1 conflates expressivity and intractability with undecidability. On finite domains (commonly assumed in all mentioned neurosymbolic frameworks) all methods are decidable.
    - C.1.2: The paper argues superiority over previous EL description fuzzy logics as it is more expressive. How is this increased expressivity used and how does this reflect in the experiments?


Minor:
- "Since $\sup$ and $\inf$ are non-differentiable: They are differentiable except when inputs are the same. They just return 0.

---

### Official Review · Reviewer_4PgB · 2025-10-30

**Soundness:** 2
**Presentation:** 3
**Contribution:** 2
**Rating:** 4
**Confidence:** 3

**Summary:**

The paper introduces GUARDNET, a neuro-symbolic framework that leverages the Guarded Fragment (GF) of first-order logic as a differentiable inductive bias for neighborhood reasoning. The authors argue that guarded quantification provides a principled locality bias, improving systematic generalization in reasoning tasks. The model extends GF with fuzzy semantics based on the Product t-norm and a Reichenbach-style implication, and integrates it with neural architectures through MLP-based predicate grounding and a hybrid domain decomposition (core and latent). Empirical results on several knowledge base completion benchmarks report large performance improvements over prior work.

**Strengths:**

The paper is well written and presents a clear exposition of the Guarded Fragment and its differentiable extension.

The idea of interpreting guarded quantification as a locality bias for reasoning over relational neighborhoods is conceptually appealing.

The hybrid domain strategy (core plus latent) is interesting and could provide a mechanism to balance logical fidelity and generalization.

**Weaknesses:**

Limited novelty of the guarded fragment focus.

Limited novelty of  fuzzy semantics and grounding.

Empirical gains lack explanatory analysis.

Missing statistical interpretation of the hybrid domain.


I will detail these points in the section below.

**Questions:**

**Limited novelty of the guarded fragment focus.**
The emphasis on the Guarded Fragment seems overstated. While the paper presents it as a novel inductive bias, similar mechanisms already appear in existing neuro-symbolic frameworks such as Logic Tensor Networks (LTN)[1] and Semantic-Based Regularization (SBR)[2]. In these systems, quantification is effectively guarded. For example,
- LTN employs diagonal quantification that limits grounding to dataset tuples, and
- SBR uses multi-sorted logic that restricts quantification domains.
- Even standard manifold regularization in extensions of SBR[3], ala "MLN smokers”, restricts reasoning to subsets defined by the manifold relation R [forall x,y: R(x,y) -> A(x) <-> A(y).

These mechanisms seem all very conceptually related to guarded quantification. The paper would have benefited from a clearer discussion of what new expressive or computational capabilities the GF formalization provides beyond these earlier works.

**Limited novelty of  fuzzy semantics and grounding.**
The proposed fuzzy semantics and neural grounding choices are conventional. The combination of Product t-norm, S-implication, and LogSumExp approximations is well known from previous differentiable fuzzy logic frameworks, e.g. LTN, logLTN. Similarly, using MLPs with concatenated embeddings to ground predicates has been standard practice even before early embedding-based approaches such as TransE and ComplEx (see corresponding related works). As presented, these sections largely restate established design patterns rather than offering conceptual innovation.  Going to simpler neural parameterization of the predicates reflects a broader and well-known trade-off in knowledge graph embeddings models. Recent approaches (for instance, those[4] using Clifford algebras for structured embeddings) explore explicit mechanisms for balancing expressivity and generalization. The paper would have benefited from positioning GUARDNET within this wider design space.

**Empirical gains lack explanatory analysis.**
The reported performance improvements are unexpectedly large given the strong methodological overlap with prior models. The paper does not isolate which components actually drive these gains. For instance, it is unclear whether the guarded restriction, the latent domain sampling, or other training details are responsible. A targeted ablation study would help validate the core claim that guardedness itself leads to better generalization.

**Statistical interpretation of the hybrid domain.**
The two-domain decomposition (core and latent) is interesting but under-specified statistically. The latent domain is sampled from a Gaussian prior, similar to variational autoencoders, yet it is not clear whether constants and latent samples share the same embedding space or semantics. There are no guarantees about model behavior outside the support of the prior (if any within). The paper should clarify how this stochastic component interacts with logical constants and what assumptions underlie their joint interpretation.


[1] Badreddine, Samy, et al. "Logic tensor networks." Artificial Intelligence 303 (2022): 103649.

[2] Diligenti, Michelangelo, Marco Gori, and Claudio Sacca. "Semantic-based regularization for learning and inference." Artificial Intelligence 244 (2017): 143-165.

[3] Marra, Giuseppe, et al. "Relational Neural Machines." ECAI 2020. IOS Press, 2020. 1340-1347.

[4] Kamdem Teyou, Louis Mozart, Caglar Demir, and Axel-Cyrille Ngonga Ngomo. "Embedding Knowledge Graphs in Degenerate Clifford Algebras." ECAI 2024. IOS Press, 2024. 1293-1300

---

### Note · Authors · 2026-01-01

**Comment:**

Dear Reviewers and AC,

Thanks for your comments and suggestions.

The Authors

**Withdrawal Confirmation:**

I have read and agree with the venue's withdrawal policy on behalf of myself and my co-authors.